# Sketched Adaptive Distributed Deep Learning: A Sharp Convergence Analysis

**Zhijie Chen**
Siebel School of Computing and Data Science
University of Illinois Urbana-Champaign
`lucmon@illinois.edu`

**Qiaobo Li**
Siebel School of Computing and Data Science
University of Illinois Urbana-Champaign
`qiaobol2@illinois.edu`

**Arindam Banerjee**
Siebel School of Computing and Data Science
University of Illinois Urbana-Champaign
`arindamb@illinois.edu`

## Abstract

Combining gradient compression with adaptive optimizers is a highly desirable goal in distributed learning, with potential benefits in both fewer communication rounds and less per-round communication. In spite of preliminary empirical promise, certain major challenges in the convergence analysis of such methods have stayed open: handling compression based approximation of both first and second moments (pre-conditioner) which appear as a ratio; avoiding dependence on the number of parameters, which is extremely large in modern deep models; and providing high-probability guarantees instead of in-expectation, which can hide high variance behavior. In this work, we introduce a family of Sketched Adaptive Distributed Learning (SADL) algorithms which can use suitable unbiased gradient sketching for compression with suitable adaptive optimization algorithms. As our main contribution, we provide theoretical convergence guarantees of SADL algorithms which addresses all of the existing challenges. In particular, our guarantees hold with high probability, picks up only a logarithmic dependence on the number of parameters, and the first and second moment approximation is handled precisely yielding a dependence on the intrinsic dimension of the loss Hessian, which is significantly smaller than the full dimensionality of deep learning models. Empirically, the SADL algorithms are shown to be competitive with and often outperform baselines on both vision and language tasks, in both supervised fine-tuning and training-from-scratch regimes. Further, the SADL algorithms are also competitive with the state-of-the-art communication-efficient distributed learning algorithms based on error feedback.

## 1 Introduction

Despite the recent progress in distributed deep learning (Liu et al., 2022), the cost of communication arguably remains the main challenge. Wang et al. (2023) showed that a 20 Gbps network bandwidth is necessary to bring the communication overhead to a suitable scale for finetuning GPT-J-6B, which is unrealistic in distributed settings. Even with good network conditions, reduction of communication complexity means one can train much larger models given the same communication budget.

The communication cost of vanilla distributed learning can be represented as $O(dT)$, where $d$ is the ambient dimension of the parameter space, i.e., the number of parameters, and $T$ is the number of communication rounds for convergence. Various methods have been proposed to minimize $T$, e.g.,

39th Conference on Neural Information Processing Systems (NeurIPS 2025).

local training (Stich, 2018), large batch training (Xu et al., 2023), etc. Folklores in centralized training regimes suggest that $T$ heavily relies on the choice of optimizers, where *adaptive methods* (such as Adam, AMSGrad, and variants) usually demonstrate faster convergence and better generalization performance, especially in transformer-based machine learning models (Reddi et al., 2019).

Alternatively, communication costs can be reduced by being more thrifty on the communication bits in each round, reducing the $O(d)$ factor to $O(b), b \ll d$. In modern deep learning models, the $O(d)$ term is the dominant factor in the communication complexity since $d \gg T$ in modern models. Considerable efforts have been devoted to design efficient *gradient compression* methods, which compress a (gradient) vector of dimension $d$ to an effective size $b$. Popular gradient compression methods include quantization (Alistarh et al., 2017; Chen et al., 2023; Reisizadeh et al., 2020; Liu et al., 2023a), sparsification (Alistarh et al., 2018; Wu et al., 2018; Rothchild et al., 2020) and sketching (Spring et al., 2019; Jiang et al., 2024; Song et al., 2023).

While progress has been made along both of these lines (see Section A), there are certain truly **unique challenges in *provably* doing both gradient compression (for communication efficiency) and adaptive optimization (for faster convergence) simultaneously**. **First**, adaptive methods (such as Adam (Kingma and Ba, 2014) and AMSGrad (Reddi et al., 2019)) work with a ratio of the first moment and second moment, and gradient compression will *approximate both the numerator and denominator of such a ratio*, making both numerical and statistical analysis of the ratio quite complex. Note that this major challenge is entirely non-existent in non-adaptive methods, which form bulk of the literature in distributed and federated learning (Rothchild et al., 2020; Shrivastava et al., 2024). **Second**, there are two primary sources of noise in any such analyses: due to mini-batching in client updates and due to stochasticity in gradient compression; and a third secondary but more complex source of noise due to sequential dependencies in adaptive updates over iterations. While an "in-expectation" analysis can mute some of these noises (Gorbunov et al., 2021), the algorithm may have high variance which will lead to unstable training in practice. To avoid such issues, our focus is to provide *high-probability convergence bounds, while taking into account all three of the above sources of noise*. **Third**, almost all existing analysis based on gradient compression picks up an *undesirable ambient dimension $d$ dependence* in the optimization analysis. Biased compressors generally need to handle errors from first-order terms because of the distortion in the gradient direction whereas unbiased compressors typically have high variance which affects the second-order terms. Since modern deep learning models have ambient dimension in hundreds of billions or even trillions, it is important to avoid such a dependence. The existing analyses on adaptive methods with gradient compression are quite alarming, which show that the iterations $T$ needed for convergence can be inversely proportional to the compression rate (Chen et al., 2022; Song et al., 2023). For constant per-round communication bits, the bounds indicate the iteration complexity $T$ to scale as $O(d)$, i.e., linearly with the ambient dimensionality, which is prohibitively large for modern deep learning models. Appendix A includes more challenges in the algorithmic and theoretical aspects.

In this work, we introduce a family of Sketched Adaptive Distributed Learning (SADL) algorithms which have the flexibility of using suitable unbiased gradient sketching compressors with suitable adaptive optimization algorithms. At a high level, SADL algorithms are analogous to previous attempts (Tang et al., 2021; Chen et al., 2022; Wang et al., 2022a), which showed preliminary empirical success of applying gradient compression with adaptive optimizers, and (Song et al., 2023) which introduced the sketch-and-desketch framework to distributed learning, but comes with a sharp convergence analysis which addresses all three of the unique challenges.

A key technical contribution of our current work is that the convergence rate of the proposed SADL algorithms depends only logarithmically (instead of linearly) on the ambient dimension $d$. The central technical challenge in addressing the dimensional dependence is to handle the entry-wise sketching noise in both the the first moments and the second moments (preconditioners) of the adaptive optimizers, which has been acknowledged to be non-trivial (Tang et al., 2021; Wang et al., 2022a). Our sharper analysis yields bounds based on the intrinsic dimension (instead of the ambient dimension $d$) of the loss Hessian in deep learning, i.e., the ratio of sum of absolute eigenvalues over the largest eigenvalue. Recent observations on the Hessian spectrum of deep learning models have demonstrated that the intrinsic dimension is significantly smaller than the ambient dimension, by showing the eigenvalues decay sharply, with most eigenvalues being close to zero (Li et al., 2020; Liao and Mahoney, 2021; Liu et al., 2023b), and even arguably conforming with a power-law decay (Xie et al., 2022; Zhang et al., 2024). This specific eigenspectrum structure provides significant advantages in the sharp analysis of sketching noise in adaptive methods. The SADL algorithms

---

**Algorithm 1** Sketched Adaptive Distributed Deep Learning (SADL)

---

**Input:** Learning rate $\eta$, initial parameters $x_0$, adaptive optimizer `ADA_OPT`
**Output:** Updated parameters $x_T$
Initialize server moments: $m_0 = 0, v_0 = 0, \hat{v}_0 = 0$, client initial parameters: $x_{0,0}^c = x_0$, client moments: $m_0^c = 0, v_0^c = 0, \hat{v}_0^c = 0, \forall c \in [C]$;
**for** $t = 1, 2, \ldots, T$ **do**
  **Client Updates:**
  **for** $c = 1, 2, \ldots, C$ **do**
    Client model synchronization: $x_{t,0}^c, m_t^c, v_t^c, \hat{v}_t^c = $ `ADA_OPT`$(x_{t-1,0}^c, m_{t-1}^c, v_{t-1}^c, \hat{v}_{t-1}^c, \bar{m}_t)$
    **for** $k = 1, 2, \ldots, K$ **do**
      Compute stochastic gradient $g_{t,k-1}^c$ with respect to the parameters $x_{t,k-1}^c$;
      Perform gradient step: $x_{t,k}^c = x_{t,k-1}^c - \eta_t g_{t,k-1}^c$;
    **end for**
    Sketch (compress) the parameter updates and send $\bar{m}_t^c$ to server: $\bar{m}_t^c = $ `sk`$(x_{t,0}^c - x_{t,K}^c)$;
  **end for**
  **Server Updates:**
  Average sketched client updates and send $\bar{m}_t$ back to clients: $\bar{m}_t = \frac{1}{C}\sum_{c=1}^C \bar{m}_t^c$;
  Update parameters and moments: $x_t, m_t, v_t, \hat{v}_t = $ `ADA_OPT`$(x_{t-1}, m_{t-1}, v_{t-1}, \hat{v}_{t-1}, \bar{m}_t)$.
**end for**

---

do not involve computing the Hessian eigenspectrum, which is only used for the convergence analysis. Further, our results do not follow from the standard toolbox for sketching methods, such as Johnson-Lindenstrauss (Kpotufe and Sriperumbudur, 2020), which ensures nearby vectors stay close after sketching, but has no direct implications in an optimization setting, especially with adaptive optimizers which will sketch both the first and second moment, and work with their ratio. Based on such analysis, our work has the following main contributions:

(1) We introduce SADL algorithms which combine random sketching and adaptive methods. While the preconditoner in adaptive methods morphs the shape of sketching noise, posing challenges in leveraging the fast-decaying Hessian eigenstructure, we prove that the proposed sketching effectively balances iteration complexity and sketching dimension $b$. We derive a high probability bound showing that a sketch dimension of $b = O(\log d)$ suffices to achieve an $\mathcal{O}(1/\sqrt{T})$ convergence rate depending only on the intrinsic dimension in non-convex deep learning settings.

(2) Unlike existing works (Reddi et al., 2020; Xie et al., 2020), we provide a general convergence analysis without assuming uniformly bounded gradient norms on either the server or client side. We prove that SADL automatically generates bounded gradients along the entire optimization trajectory, where the involvement of mini-batch stochasticity and multiple local training steps calls for careful analysis on connecting the noisy local training steps with the global loss.

(3) We validate our theoretical claims with empirical evidence on deep learning models from vision (ResNet, Vision Transformer) and language (BERT) tasks. We cover both fine-tuning and training-from-scratch regimes. SADL in general achieves comparable performance with adaptive methods without any gradient compression, using only $0.1\%$ of the full dimsioanlity. Further, SADL is competitive with the state-of-the-art algorithms based on error feedback and adaptive methods.

## 2 Sketched Adaptive Distributed Deep Learning

A canonical distributed learning setting involves $C$ clients, each associated with a local data distribution $\mathcal{D}_c$. The goal is to minimize the averaged empirical risk: $\mathcal{L}(x) = \frac{1}{C}\sum_{c=1}^C \mathbb{E}_{\xi \sim \mathcal{D}_c} l(x, \xi)$, where $l$ is the loss function, $x \in \mathbb{R}^d$ is the parameter vector, and $\xi$ is the data sample. We denote $\mathcal{L}^c(x) = \mathbb{E}_{\xi \sim \mathcal{D}_c} l(x, \xi), c \in [C]$ as the client loss computed over the local distribution. We denote $g_{t,k}^c$ as the mini-batch gradient over $\mathcal{L}^c(x)$ at global step $t$ and local step $k$.

Algorithm 1 presents a generic framework of communication-efficient adaptive methods, which calls adaptive optimizers as subroutines. We denote $T$ as the total training rounds. At each round, after $K$ local SGD steps, client $c$ sends to the server the sketched local model updates with a sketching operator `sk`: $\mathbb{R}^d \mapsto \mathbb{R}^b$. If $b \ll d$ without deteriorating the performance too much, the communication

**Algorithm 2** `ADA_OPT` (AMSGrad)

---

**Input:** iterate $x_{t-1}$, moments $m_{t-1}, v_{t-1}, \hat{v}_{t-1}$, sketched updates $\bar{m}_t$

**Parameters:** Learning rate $\kappa$, $\beta_1$, $\beta_2$, Small constant $\epsilon$, Sketch size $b$

**Output:** Updated parameters $x_t$, and moments $m_t, v_t, \hat{v}_t$

Update $\quad m_t = \beta_1 \cdot m_{t-1} + (1 - \beta_1) \cdot \texttt{desk}((\bar{m}_t)_{1:\lfloor b/2 \rfloor})$;

Update $\quad v_t = \beta_2 \cdot v_{t-1} + (1 - \beta_2) \cdot \texttt{desk}((\bar{m}_t)_{\lfloor b/2 \rfloor : b})^2$;

Update $\quad \hat{v}_t = \max(\hat{v}_{t-1}, v_t)$.

Update $\quad x_{t+1} = x_t - \frac{\kappa}{\sqrt{\hat{v}_t} + \epsilon} \cdot m_t := x_t - \kappa \hat{V}_t^{-1/2} m_t$.

---

cost per round can be reduced from $\mathcal{O}(d)$ to $\mathcal{O}(b)$. Algorithm 2 projects the compressed updates and second moments back to the ambient dimension using a desketching operator $\texttt{desk}: \mathbb{R}^b \mapsto \mathbb{R}^d$ and implements a single-step adaptive optimization. The server and clients call Algorithm 2 at every epoch, i.e. communication round, to update the global model and synchronize local models. Notice that in Algorithm 1, both the clients and the server transmit sketched vectors which results in an overall $\mathcal{O}(bT)$ communication cost.

**Remark 1.** *(Sketching Randomness)* At each single round, the sketching operators $\texttt{sk}$'s are shared among clients, via the same random seed. On the other hand, we use fresh $\texttt{sk}$'s at different rounds to get statistical independence which helps the analysis. $\qquad\square$

We list some desired properties for $\texttt{sk}, \texttt{desk}$ and discuss sketching methods which satisfy them.

**Property 1.** *(Linearity).* *The compression operators are linear w.r.t the input vectors, i.e.* $\texttt{sk}(\sum_{i=1}^n v_i) = \sum_{i=1}^n \texttt{sk}(v_i)$ *and* $\texttt{desk}(\sum_{i=1}^n \bar{v}_i) = \sum_{i=1}^n \texttt{desk}(\bar{v}_i)$, $\forall \{v_i, \bar{v}_i \in \mathbb{R}^d\}_{i=1}^n$.

**Property 2.** *(Unbiased Estimation).* *For any vector* $v \in \mathbb{R}^d$, $\mathbb{E}[\texttt{desk}(\texttt{sk}(v))] = v$.

**Property 3.** *(Bounded Vector Products).* *For any fixed vector* $v, h \in \mathbb{R}^d$, $\mathbb{P}(|\langle \texttt{desk}(\texttt{sk}(v)), h \rangle - \langle v, h \rangle| \geq (\frac{\log^{1.5}(d/\delta)}{\sqrt{b}})\|v\|\|h\|) \leq \Theta(\delta)$.

We denote $R \in \mathbb{R}^{b \times d}$ as the sketching operator, and $\texttt{sk}(v) = Rv$ and $\texttt{desk}(\bar{v}) = R^\top \bar{v}$. Different instantiations of $R$ constitute a rich family of sketching operators, including i.i.d. isotropic Gaussian (Song et al., 2023), Subsampled Randomized Hadamard Transform (SRHT) (Lu et al., 2013), and Count-Sketch (Charikar et al., 2002), among others. The specific bounds for these special cases can be found in Appendices B.1, B.2, and B.3 respectively.

## 2.1 Convergence Analysis

We first state a set of standard assumptions commonly used in the literature of first-order stochastic methods. We will use $\|\cdot\|$ to denote $L_2$-norm throughout the work.

**Assumption 1.** *(Bounded Global Gradients). Square norm of the gradient is uniformly bounded, i.e.,* $\|\nabla \mathcal{L}(x)\|^2 \leq G_g^2$.

**Assumption 2.** *(Bounded Client Gradients). For every client, there exists a constant* $G_c \geq 0$, *such that* $\|\nabla \mathcal{L}^c(x)\|^2 \leq G_c^2$, $c \in [C]$.

For simplicity, in this section we define $G := \max\{\max\{G_c\}_{c=1}^C, G_g\}$. **In Section 3, we show that Assumptions 1 and 2 are not necessary to derive the convergence bound we present.** We assume the local stochastic mini-batch noise is sub-Gaussian, which is widely adopted in first-order optimization (Harvey et al., 2019; Mou et al., 2020).

**Assumption 3.** *(Sub-Gaussian Noise). The stochastic noise* $\|\nabla \mathcal{L}^c(x) - g^c(x)\|$ *at each client is a* $\sigma$*-sub-Gaussian random variable, i.e.* $\mathbb{P}(\|\nabla \mathcal{L}^c(x) - g^c(x)\| \geq t) \leq 2\exp(-t^2/\sigma^2)$, *for all* $t \geq 0$.

Besides, we present assumptions on the Hessian eigenspectrum $\{\lambda_i, v_i\}_{i=1}^d$ of the loss function $\mathcal{L}$.

**Assumption 4.** *(Hessian Matrix Eigenspectrum) The smoothness of the client loss function* $\mathcal{L}_i$, *i.e. the largest eigenvalue of the loss Hessian* $H_{\mathcal{L}_i}$ *is bounded by* $L$.

The local smoothness assumption is commonly used in distributed learning settings (Safaryan et al., 2021; Fatkhullin et al., 2024) and holds for general deep learning losses. It can be directly derived from Assumption 4 that the global loss $\mathcal{L} = \frac{1}{C}\sum_{c=1}^C \mathcal{L}_c$ is $L-$smooth.

**Definition 2.1.** *(Intrinsic Dimension) Let $\{\lambda_i\}_{i=1}^d$ be the eigenspectrum of the loss Hessian $H_{\mathcal{L}}$. The intrinsic dimension is defined as $\mathcal{I} = \sum_{i=1}^d |\lambda_i| / \max_i |\lambda_i|$.*

The definition of intrinsic dimension is analogous to that in the classical literature (Ipsen and Saibaba, 2024), but we take the absolute values of eigenvalues. Intuitively, the Hessian matrix possesses an **anisotropic** structure in different directions, whereas the conventional smoothness is a pessimistic estimation of the loss curvature. A large volume of recent empirical literature has identified that the intrinsic dimension of the Hessian in deep learning losses is significantly smaller than the ambient dimensionality $d$. (Ghorbani et al., 2019; Li et al., 2020; Liu et al., 2023b) show the eigenspectrum enjoys a sharp decay in magnitude. (Sagun et al., 2016; Liao and Mahoney, 2021) show the eigenspectrum have bulk parts concentrate at zero. (Xie et al., 2022; Zhang et al., 2024) further show the eigenvalues satisfy a power-law distribution, and in this case the intrinsic dimension is a constant independent of $d$. We quote their plots in Appendix E for completeness. Our empirical verification under the setting of distributed learning can also be found in Fig. 5 in Appendix E.

**Remark 2.** *(Three types of noises in Algorithm 1).* One of the key technical contributions of this work is to theoretically balance the noises from different sources. The noise in the training process stems from the local mini-batch training, the compression error due to sketching, and the aggregate noise over the training horizon. The mini-batch stochastic error is $\sigma$-sub-Gaussian by Assumption 3. The sketching error depends on the specific choice of sketching methods, but is always subject to the bounded property on vector products (Property 3). We will denote the 'bad probability' from mini-batch and sketch respectively as $\delta_g$ and $\delta$, which are usually viewed as a tiny value ($10^{-5}$) in high probability bounds. These two kinds of noise are unbiased, but leads to sequential dependencies. In the analysis (Appendix C), we will show that the aggregated noise due to the sequential dependencies form a martingale, which will be used to derive high-probability concentration bounds. We denote $\nu$ as the scale of the $\psi_2$-norm (Vershynin, 2018) in the martingale. □

Now we characterize the convergence of Algorithm 1 in Theorem 2.2. All technical proofs for this section are in Appendix C and we provide an outline of the proof techniques in Section 2.2.

**Theorem 2.2.** *(Informal version of Theorem C.1) Suppose the sequence of iterates $\{x_t\}_{t=1}^T$ is generated by Algorithm 1 with a constant learning rate $\eta_t \equiv \eta$. Under Assumptions 1-4, for any $T$ and $\epsilon > 0$, with probability $1 - \Theta(\delta)$,*

$$\frac{\kappa \eta K}{\eta K G \sqrt{J(b,d)} + \epsilon} \sum_{t=1}^T \|\nabla \mathcal{L}(x_t)\|^2 \leq \mathcal{L}(z_1) + \eta \sqrt{T} \mathcal{O}(1 + J(b,d)) + \eta^2 T \mathcal{O}(1 + \mathcal{I}(1 + J(b,d))^2),$$

*where $J(b,d) = \frac{\log^{1.5}(CKd^2T^2/\delta)}{\sqrt{b}}$, and $z_1 = \frac{1}{1-\beta_1}x_1 - \frac{\beta_1}{1-\beta_1}x_0$.*

**Remark 3.** *(Nature of High Probability)* While our convergence result holds with high probability, the informal version in Theorem 2.2 hides the dependencies on the three types of noise (Remark 2). The dependencies follow exponential concentrations, and are shown in Corollaries 1 and 2. □

**Remark 4.** *(Rate of Convergence)* The rate of convergence may be hard to read off from Theorem 2.2. At a high level, we show in Corollaries 1 and 2 that the rate is $O(1/T)$ when $T \leq O(1/\epsilon^2)$ and is $O(1/\sqrt{T})$ when $T \geq \Omega(1/\epsilon^2)$. The specific results provide additional details. □

A non-asymptotic convergence bound of training with practical decaying learning rates can be found in Theorem C.4 in appendix. Given that we only introduce logarithmic factors on $d$ in the iteration complexity and the per-round communication $b$ is a constant, the total communication bits of training a deep model till convergence is also logarithmic w.r.t $d$.

To better understand Theorem 2.2, we can investigate different regimes based on the training stages. For large $T$, where $T \geq \Omega(1/\epsilon^2)$, we can achieve an $\mathcal{O}(1/\sqrt{T})$ convergence rate in Corollary 1.

**Corollary 1.** *(Asymptotic Regime of Theorem 2.2) With the same condition as in Theorem 2.2 and a constant learning rate $\eta_t \equiv \frac{1}{K\sqrt{T}}$, for sufficiently large $T \geq \frac{G^2}{\epsilon^2}$, with probability $1 - \Theta(\delta) - \mathcal{O}(\exp(-\Omega(\nu^2))) - \delta_g$,*

$$\frac{1}{T}\sum_{t=1}^T \|\nabla \mathcal{L}(x_t)\|^2 \leq \frac{4}{\sqrt{T}}(1 + J_2)^2 \frac{4\kappa \mathcal{I} L + G}{(1-\beta_1)^2} \frac{G^2}{\epsilon} + \frac{2\mathcal{L}(z_1)\epsilon}{\kappa\sqrt{T}} + \frac{2}{\epsilon}\frac{LG^2}{\sqrt{T}} + \nu\frac{2}{\sqrt{T}}(J_2 G^2 + \sigma \log^{\frac{1}{2}}(\frac{2T}{\delta_g})),$$

*where $\delta, \delta_g$ and $\nu$ correspond to the three noise types (Remark 2), and $J_2 := \frac{\log^{1.5}(CKdT^2/\delta)}{\sqrt{b}}$.*

More interestingly, when $T$ is relatively small, i.e., $T \leq O(1/\epsilon^2)$, we can observe that the coefficient of $\|\nabla \mathcal{L}(x_t)\|^2$ on the left hand side in Theorem 2.2 and C.4 is approximately a constant, given that $\epsilon$ is relatively small. Therefore, SADL can achieve an $\mathcal{O}(1/T)$ convergence for small $T$, which yields faster convergence rate than non-adaptive methods.

**Corollary 2.** *(Near-initialization Regime of Theorem 2.2) With the same condition as in Thereom 2.2 and a constant learning rate $\eta_t \equiv \frac{1}{K\sqrt{T}}$, set $b \geq \log^3(CKd^2T^2/\delta)$ and constant $J_3 > \sqrt{2}G$, then for any $T \leq \frac{J_3 - \sqrt{2}G}{\epsilon^2}$, with probability $1 - \Theta(\delta) - \mathcal{O}(\exp(-\Omega(\nu^2))) - \delta_g$,*

$$\frac{1}{J_3 T} \sum_{t=1}^{T} \|\nabla \mathcal{L}(x_t)\|^2 \leq \frac{\mathcal{L}(z_1)\epsilon}{\kappa T} + \frac{1}{\epsilon} \frac{LG^2}{T} + \frac{\nu}{T}(G^2 + \sigma \log^{\frac{1}{2}}(\frac{2T}{\delta_g})) + \frac{8}{T} \frac{4\kappa \mathcal{I} L + G}{(1-\beta_1)^2} \frac{G^2}{\epsilon},$$

*where $\delta, \delta_g$ and $\nu$ correspond to the three noise types (Remark 2).*

## 2.2 Technical Details and Proof Sketch

In this section, we provide a proof sketch behind the main results. We focus on the proof of Theorem 2.2, and the proof of Theorem C.4 shares the main structure. The proof contains several critical components, which are unique to adaptive methods. We adopt AMSGrad (Alg. 2) as the server optimizer and it would be straightforward to extend the analysis to other adaptive methods.

First, we introduce the descent lemma for AMSGrad. For conciseness, we denote the precondtioner matrix $\mathrm{diag}((\sqrt{\hat{v}_t} + \epsilon)^2)$ as $\hat{V}_t$. Define an auxiliary variable $z_t = x_t + \frac{\beta_1}{1-\beta_1}(x_t - x_{t-1})$. The trajectory of $\mathcal{L}$ over $\{z_t\}_{t=1}^T$ can be tracked by the following lemma.

**Lemma 2.3.** *(Informal version of Lemma C.2) For any step $t \in [T]$,*

$$\mathcal{L}(z_{t+1}) \lesssim \mathcal{L}(z_t) - \frac{\kappa \eta}{C} \sum_{c=1}^{C} \sum_{k=1}^{K} \nabla \mathcal{L}(x_t)^\top \hat{V}_{t-1}^{-1/2} R_t^\top R_t g_{t,k}^c + (z_t - x_t)^\top H_{\mathcal{L}}(\hat{z}_t)(z_{t+1} - z_t),$$

*where $H_{\mathcal{L}}(\hat{z}_t)$ is the loss Hessian at some $\hat{z}_t$ within the element-wise interval of $[x_t, z_t]$, and $\lesssim$ omits the less important terms.*

Our objective henceforth is to bound the first-order descent term and the second-order quadratic term on the right hand side respectively.

**Second-Order Quadratic Term.** Denote $\{\lambda_j, v_j\}_{j=1}^d$ as the eigen-pairs of $H_{\mathcal{L}}(\hat{z}_t)$. The quadratic term can be written as $(z_t - x_t)^\top H_{\mathcal{L}}(\hat{z}_t)(z_{t+1} - z_t) = \sum_{j=1}^d \lambda_j \langle z_{t+1} - z_t, v_j \rangle \langle z_t - x_t, v_j \rangle$. The inner product term is a projection of the updates onto anisotropic bases. Since $z_{t+1} - z_t$ and $z_t - x_t$ can be expressed by $x_{t+1} - x_t$ and $x_t - x_{t-1}$, we can bound the quadratic term using the following lemma.

**Lemma 2.4.** *For any $t \in [T]$, $|\langle x_t - x_{t-1}, v_j \rangle| \leq \kappa \eta (1 + \frac{\log^{1.5}(CKtd/\delta)}{\sqrt{b}}) \frac{KG}{\epsilon}$, with probability $1 - \delta$.*

A proof of a generalized version of this statement is deferred to the appendix. Induction method is used to address the temporal dependence introduced by the momentum factor in AMSGrad. Combining Lemma 2.4 with Assumption 4 yields a dimension-free bound on the second-order quadratic term.

**Remark 5.** A straightforward application of smoothness to the second-order term yields a quadratic term $\|R^\top Rg\|^2$, which is linearly proportional to $d$ in scale (Rothchild et al., 2020; Song et al., 2023). We avoid this dimension dependence by combining Property 3 of sketching and the intrinsic dimension of the deep learning Hessian matrix. □

**First-Order Descent Term**. The first-order term in the descent lemma can be decomposed into three components, which we will handle separately:

$$\nabla \mathcal{L}(x_t)^\top \hat{V}_{t-1}^{-1/2} R_t^\top R_t g_{t,k}^c = \underbrace{\nabla \mathcal{L}(x_t)^\top \hat{V}_{t-1}^{-1/2} \nabla \mathcal{L}^c(x_t)}_{\mathcal{D}_1^c} + \underbrace{\nabla \mathcal{L}(x_t)^\top \hat{V}_{t-1}^{-1/2}(R_t^\top R_t g_{t,k}^c - \nabla \mathcal{L}^c(x_{t,k}^c))}_{\mathcal{D}_2^c}$$
$$+ \underbrace{\nabla \mathcal{L}(x_t)^\top \hat{V}_{t-1}^{-1/2}(\nabla \mathcal{L}^c(x_{t,k}^c) - \nabla \mathcal{L}^c(x_t))}_{\mathcal{D}_3^c}.$$

First, $\mathcal{D}_3^c$ can be reduced to a second-order term by Taylor expansion on $\nabla \mathcal{L}$. Since this term does not involve any stochasticity from random sketching, we can directly upper bound $\mathcal{D}_3^c$ by Cauchy-Schwartz. Next, since $\frac{1}{C} \sum_{c=1}^C \nabla \mathcal{L}^c(x_t) = \nabla \mathcal{L}(x_t)$, $\mathcal{D}_1^c$ can be viewed as a scaled squared gradient norm. Applying element-wise high probability bound on random sketching yields a lower bound.

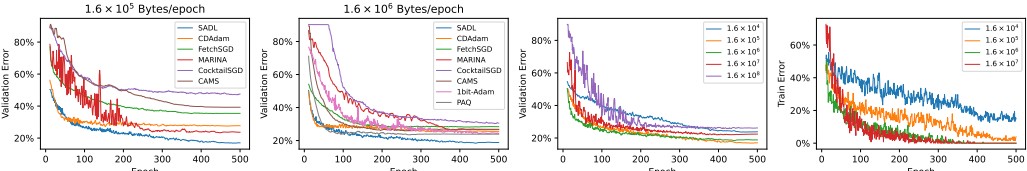

Figure 1: Model performance on CIFAR-10 with ResNet of 42M parameters. The plot starts from the 10th epoch for better demonstration; First (Second): Validation errors under compression rate $0.1\%$ ($1\%$). Third: Validation error on SADL with different communication costs. The legend $1.6 \times 10^8$ represents training in the ambient dimension without sketching. Fourth: Training error on SADL with different communication costs. Higher compression rate improves the convergence rate in training and the peak validation error is achieved when the compression rate is $0.1\%$.

**Lemma 2.5.** *For $\hat{V}_{t-1}$ generated by Algorithm 1 (SADL), with probability $1 - \delta$,*

$$\nabla \mathcal{L}(x_t)^\top \hat{V}_{t-1}^{-1/2} \nabla \mathcal{L}(x_t) \geq M^{-1} \|\nabla \mathcal{L}(x_t)\|^2,$$

*where $M = \sqrt{1 + \frac{\log^{1.5}(CKtd^2)}{\sqrt{b}} \eta KG} + \epsilon$.*

**Martingale for zero-centered noise.** $\mathcal{D}_2^c$ contains a zero-centered noise term $R_t^\top R_t g_{t,k}^c - \nabla \mathcal{L}^c(x_{t,k}^c)$, where the randomness is over $R_t$ and the mini-batch noise at round $t$. Although $x_{t,k}^c$ has temporal dependence, the fresh noise due to mini-batching and sketching-desketching at round $t$ is independent of the randomness in the previous iterations. Therefore, the random process defined by the aggregation of $\mathcal{D}_2^c$ forms a martingale. The martingale difference can be bounded with high probability under our proposed sketching method. By adapting Azuma's inequality on a sub-Gaussian martingale, we have

**Lemma 2.6.** *With probability $1 - \mathcal{O}(\exp(-\Omega(\nu^2))) - \delta - \delta_g$,*

$$\sum_{t=1}^{T} \left| \frac{1}{C} \sum_{c=1}^{C} \sum_{k=1}^{K} \nabla \mathcal{L}(x_t)^\top \hat{V}_{t-1}^{-1/2} (R_t^\top R_t g_{t,k}^c - \nabla \mathcal{L}^c(x_{t,k}^c)) \right| \leq \nu K \sqrt{T} \left( \frac{\log^{1.5}(CKTd/\delta)}{\sqrt{b}} \frac{G^2}{\epsilon} + \frac{\sigma}{\epsilon} \log^{\frac{1}{2}} (\frac{2T}{\delta_g}) \right).$$

Finally, applying union bounds to these parts and telescoping the descent lemma leads to Theorem 2.2.

**Remark 6.** While the intrinsic dimension plays a key role in avoiding the ambient dimension dependence in our convergence bound, it alone is insufficient to address all the technical challenges in sketched adaptive distributed learning. In fact, intrinsic dimension only addresses the second-order term in optimization. The analysis of sufficient descent (Lemma 2.3 and 2.5) and bounded aggregate sketching noise (Lemma 2.6) manages to address the noise, the norm of which is in-expectation linear to $d$, in the first-order term. $\qquad \square$

## 3 Bounded Gradient Norm Along Optimization Trajectory

Although the gradient norm assumptions (Assumption 1 and 2) are standard in adaptive optimization (Reddi et al., 2020) and distributed learning research (Basu et al., 2019; Xie et al., 2020), in this section, we show that the two assumptions are not necessary to derive the convergence bound.

Our main idea is to show the gradient norm is bounded over the entire optimization path with high probability. We rely on the following lemma to demonstrate the boundedness.

**Lemma 3.1.** *For any $L$-smooth function $\mathcal{L}(x)$ with optimal value $\mathcal{L}^* \geq 0$, $\|\nabla \mathcal{L}(x)\|^2 \leq 2L\mathcal{L}(x)$.*

As stated in Lemma 3.1, for any smooth function, the gradient norm can be bounded by the function value at the specific iterate. That being said, we can derive an upper bound on the gradient norm along the optimization trajectory via bounding the function values over the iterates. However, the technical difficulty of the analysis lies in the involvement of the local training steps, which might be noisy and the relation of which with the global iterate is unclear.

Our analysis can be divided into two steps: 1) We first relate the averaged local gradient norm to the global function value based on the local smoothness. Notice that this step does not require any additional assumptions, such as the deviation between local and global function values; 2) We apply

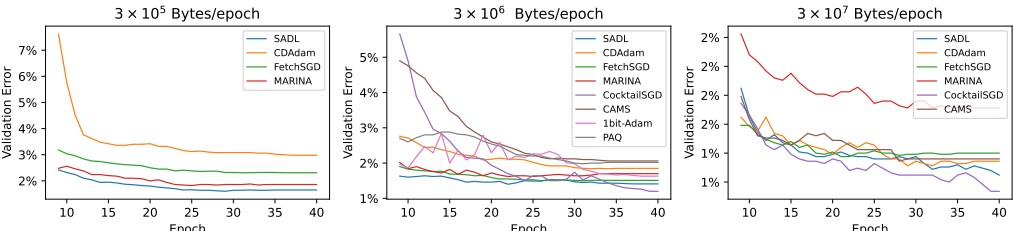

Figure 2: Validation Error on CIFAR-10. We finetune a ViT-base model (with 86M parameters). 1 bit-Adam has comparable communication cost with $3 \times 10^6$ Bytes/epoch. SADL shows competitive performance under all communication budgets.

the induction method to show the global loss is contained in the neighborhood of the function value at initialization, and the bound of the gradient norm follows immediately by applying Lemma 3.1.

The following lemma shows how the local gradient norm can be related to the global loss at iterate $x_t$, which will be a key component in the following analysis.

**Lemma 3.2.** *Under Assumption 3, Let $\eta \leq \frac{1}{2L\sqrt{K}}$. The local gradients as of $k \leq K$ can be bounded with probability $1 - CK\delta_c - CK \exp(-\Delta^2/\sigma^2)$ by*

$$\frac{1}{C} \sum_{c=1}^{C} \|\nabla\mathcal{L}^c(x_{t,\tau}^c)\| \leq 2\sqrt{L}\sqrt{\mathcal{L}(x_t)} + 2\sqrt{2\Delta^2 \ln \frac{2}{\delta_c}} + \Delta.$$

Next, we show the global loss is contained in the neighborhood of the function value at initialization. With a slight abuse of notation, we define a constant $G := \max\{2\Delta^2(1 + \sqrt{2\ln \frac{2CK}{\delta_c}})^2, \tilde{O}(1)/\sqrt{b} + \mathcal{M}\}$ where the full form can be found in (7) in Appendix D and the following lemma holds.

**Lemma 3.3.** *We can derive an upper bound on $\mathcal{L}(z_T)$ with high probability,*

$$\mathcal{L}(z_T) \leq \kappa\eta\sqrt{T}\mathcal{M}_1 G + \kappa\eta\sqrt{T}\mathcal{M}_2\sqrt{G} + \mathcal{M}_3 + \sum_{t=1}^{T-1}(\kappa\eta^2\mathcal{M}_4 G^{3/2} + \kappa\eta^2\mathcal{M}_5 G + \kappa\eta^2\mathcal{M}_6\sqrt{G} + \kappa^2\eta^2\mathcal{M}_7 G),$$

*where $\{\mathcal{M}_i\}_{i=1}^7$ are constants independent of $\kappa, \eta$ and $G$, defined in Appendix D.*

With the closeness of $z_T$ and $x_T$, we can show $\mathcal{L}(x_T) \leq \frac{G}{2L}$ under appropriate choice of $\kappa$ and $\eta$, and it is sufficient to ensure the gradient norm is bounded over the entire optimization path, which directly leads to the following convergence guarantee. The full proof can be found in Appendix D.

**Theorem 3.4.** *Let $\eta = \frac{\eta_0}{\sqrt{T}}$ subject to $\eta_0 \leq \min\{\frac{\epsilon}{6\sqrt{L}}(1 + \frac{\log^{1.5}(CKTd^2/\delta)}{\sqrt{b}})^{-1}, \frac{\sqrt{T}}{2L\sqrt{K}}\}$ and $\kappa = \frac{1}{\sqrt{G}}$. Then under Assumption 3 and 4, with probability $1 - T\exp(-\Omega(\nu^2)) - TCK\delta_c - TCK\exp(-\Delta^2/\sigma^2) - T\delta$, the gradient on the iterates $x_t$ generated by Algorithm 1 are bounded by $G$, i.e. $\|\nabla\mathcal{L}(x_t)\|^2 \leq G$. Consequently, the averaged gradient converges with rate $O(1/\sqrt{T})$ by*

$$\frac{1}{T} \sum_{t=1}^{T} \|\nabla\mathcal{L}(x_t)\|^2 \leq \frac{G}{2\kappa\eta_0 L^2\sqrt{T}}(\eta K\mathcal{M}_8 + \epsilon),$$

*where $\mathcal{M}_8 := \sqrt{1 + \frac{\log^{1.5}(CKd^2T^2/\delta)}{\sqrt{b}}}(\sqrt{2G} + 2\Delta(1 + \sqrt{2\ln \frac{2}{\delta_c}}))$.*

## 4 Empirical Studies

In this section, we instantiate the algorithm framework of SADL to demonstrate the effect of sketching in common distributed deep learning settings.

**Experimental Configurations.** We adopt three experimental settings, from vision to language tasks. For the vision task, we train a ResNet101 (Wu and He, 2018) with a total of 42M parameters from scratch and finetune a ViT-Base (Dosovitskiy et al., 2020) with 86M parameters on CIFAR-10 (Krizhevsky et al., 2009). For the language task, we adopt SST2, a text classification task, from

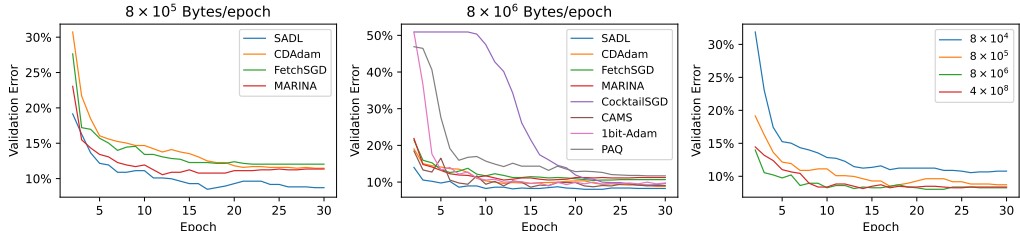

Figure 3: Validation Error on SST2 (GLUE) with BERT of 100M parameters. Left: compression rate $0.2\%$; Middle: $2\%$; Right: SADL with communication costs $\{8 \times 10^4, 8 \times 10^5, 8 \times 10^6, 4 \times 10^8\}$ Bytes/epoch. The legend $4 \times 10^8$ represents training in the ambient dimension without sketching. Higher compression rate improves the convergence rate and all compression rates achieve comparable test errors at the end of training.

the GLUE benchmark (Wang et al., 2018). We train a BERT model (Devlin, 2018) with 100M parameters. For all experiments we split the training dataset uniformly over 5 clients. Our baselines include FetchSGD (Rothchild et al., 2020), MARINA (Gorbunov et al., 2021), CocktailSGD (Wang et al., 2023), CDAdam (Wang et al., 2022a), 1 bit-Adam (Tang et al., 2021), FedCAMS (Wang et al., 2022b) and FedPAQ (Reisizadeh et al., 2020). A comparison of the theoretical guarantees of the baselines can be found in Table 12 in the Appendix. We define the compression rate as the ratio between the transmission size and the full model size (both in bytes). Specifically, for SADL, the compression rate is $b/d$, i.e. the ratio between sketch size and full dimensionality.

In each set of experiments, we compare SADL and baseline algorithms under compression rates of $0.1\%$ and $1\%$ respectively. For fair comparison, All algorithms, including SADL, use Adam as the server optimizer when applicable. SADL adopts SRHT as the sketching algorithm. Notice that most existing works only demonstrate the performance under up to $1\%$ compression rate, while our work further pushes the limit of compression to $0.1\%$. We will show SADL's highly competitive performance in this extreme case, where some baselines fall short.

**Sharp-Decaying Hessian Eigenspectrum**. Our theoretical result builds upon the notion of intrinsic dimension. While existing research has repeatedly shown supporting evidence on the sharp-decaying eigenspectrum, we also provide a verification in distributed deep learning in Fig. 5 in the Appendix.

**Convergence Results.** We present the empirical performance of SADL under three different tasks and report the mean values across 3 independent runs, with sources of randomness including client data partitioning, data shuffling, and sketching. The statistical significance analysis and the hyper-parameter budgets can be found in Appendix E. Fig. 1 shows the error curves on the validation set of CIFAR-10 when training ResNet(40M) with communication budget in $\{1.6 \times 10^5, 1.6 \times 10^6\}$ bytes/epoch, representing $0.1\%$ and $1\%$ compression rate respectively. For SADL, the two compression rates are achieved with sketch size $b \in \{4 \times 10^4, 4 \times 10^5\}$. The compression rate of 1bit-Adam and FedPAQ is $3\%$, and are compared with other baselines with compression rate $1\%$. We can observe that under both communication budgets, SADL outperforms other optimizers by a significant margin. Further, we compare the model performance of SADL with different sketch sizes and find that in this experimental setting, the validation error is not monotonic with the sketch size and reaches the peak value when $b = 4 \times 10^4$ (i.e. compression rate $0.1\%$). On the other hand, the training error, which better reflects the convergence speed, is strictly monotonic with sketch sizes – larger sketch size leads to faster convergence and agrees with our theory. The discrepancy between the two rates suggests that sketching methods may implicitly improve the model generalization ability.

Similar phenomenon is observed in the language task. Fig. 3 shows the validation errors of training SST2 with BERT (100M parameters). The compression rates are selected from $\{0.2\%, 2\%\}$. Fig. 3 includes baseline methods which achieve comparable performance. The full plot including all baselines is presented in Fig. 7 in the appendix. We observe SADL converges faster and achieves slightly better validation performance across communication budgets. In Fig. 3, we further compare SADL with different compression rates from $\{0.02\%, 0.2\%, 2\%\}$ and distributed adaptive method without sketching. Remarkably, 1) SADL converges faster with higher compression rates; 2) Under compression rate $0.2\%$, SADL achieves almost the same performance as the original unsketched version; 3) Given that the ambient dimension is 100M, it is thrilling to see under an extremely low compression rate ($0.02\%$), SADL still achieves comparable performance as trained in the ambient dimension. We also present results from finetuning a ViT-Base model (80M parameters) in Fig. 2.

Table 1: Ablation Study of SADL on the ResNet experiment with communication costs $\{1.6 \times 10^5, 1.6 \times 10^6\}$ Bytes/epoch. Validation errors of the last communication round are displayed. The comparison is across different combinations of server optimizers {Adam. AMSGrad} and sketching algorithm {SRHT, CountSketch}. Adam + SRHT is the recommended combination in practice.

|  | $1.6 \times 10^5$ Bytes/epoch | | $1.6 \times 10^6$ Bytes/epoch | |
|  | SRHT | CountSketch | SRHT | CountSketch |
| --- | --- | --- | --- | --- |
| Adam | 17.0% | 23.8% | 16.7% | 23.6% |
| AMSGrad | 18.1% | 25.1% | 17.0% | 24.7% |

The compression rates are selected from $\{0.1\%, 1\%, 10\%\}$. The full plot can be found in Fig. 6 in the appendix. We observe that SADL is advantageous especially under lower compression rates. Remarkably, CocktailSGD achieves the best validation error under compression rates $1\%$ and $10\%$, while falling short in $0.1\%$. SADL has a consistent performance across all communication budgets.

**Performance under Extremely Low Communication.** We experiment with extremely low compression rates to show the theoretical logarithmic dependence can be empirically grounded. The experiments are conducted under the same setting as Figure 1 and 3 respectively. We adopt compression rates from 0.001% to 1% where the lowest amounts to $b = 400$ in the ResNet experiment and $b = 2000$ in the SST-2 task. We present the validation errors along the training process in Figure 8 in Appendix E. We observe that although the validation accuracies converge to distinct values, SADL converges under all sketch sizes. More interestingly, different sketch sizes achieve similar convergence speed. Even under extremely tiny sketch sizes, SADL converges in the first 100 (25 *resp.*) epochs in CIFAR-10 (SST2 *resp.*) task. This observation aligns with our theoretical results on the logarithmic dependence on $d$ in the convergence rate.

**Ablation Studies.** SADL naturally supports a custom choice of server optimizers and sketching algorithms in practice. In the previous experiments, we choose Adam as the server optimizer and SRHT as the sketching algorithm. Since the mainstream adaptive optimizers share similar designs in utilizing momentum and preconditioners, the theory we developed can be easily extended to Adam. The theoretical guarantees can also be seamlessly adapted to other sketching algorithms. We extend our experimental results to the combination of [Adam, AMSGrad] $\times$ [SRHT, CountSketch]. Table 1 presents the validation errors during the last communication round.

Regarding the choice of server optimizer, Adam achieves slightly better performance than AMSGrad. Regarding the choice of sketching methods, SRHT consistently outperforms CountSketch, while the latter is still competitive with other baseline methods in Fig. 1. This observation aligns with the error bound of CountSketch (Lemma B.3), which differs by a $1/\sqrt{b}$ factor with SRHT (Lemma B.1).

## 5 Conclusion

In this paper, we investigated sketched adaptive methods for distributed learning. While the motivation behind combining sketching and adaptive methods is clear, there is limited understanding on its empirical success. We show an exponential improvement in the communication cost from $O(d)$ to $O(\log d)$, i.e., logarithmic in the number of parameters. The improvement is especially important for modern deep learning models with large number of parameters. Our analysis introduces a novel technique that handles three sources of noise and adaptive optimization, both of which pose key challenges in applying sketching methods to communication-efficient adaptive distributed deep learning. We empirically show that the proposed SADL consistently outperforms the existing baselines, meaning our method is practical, while having strong theoretical guarantees.

## Acknowledgment

The work was supported by the National Science Foundation (NSF) through awards IIS 21-31335, OAC 21-30835, DBI 20-21898, as well as a C3.ai research award. Compute support for the work was provided by the National Center for Supercomputing Applications (NCSA) and the Illinois Campus Cluster Program (ICCP).

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

# A Related Work and Key Challenges

In this section, we briefly discuss the related work especially in the context of the key challenges in combining gradient sketching with adaptive methods. At a high level, analysis of adaptive methods with gradient compression has to manage the error or variance due to compression for both the first moment and second moment terms, which appear as a ratio in the algorithm. There are two broad categories of gradient compressors: biased and unbiased. Biased compressors generally need to handle errors from first-order terms because of the distortion in the gradient direction whereas unbiased compressors typically have high variance which affects the second-order terms.

**Unbiased Compression.** Denote $\mathcal{C}$ as the compression operator over (gradient) vector $x$. The compression error $\omega$ can be characterized by $\mathbb{E}\|\mathcal{C}(x) - x\| \leq \omega\|x\|$. The convergence rates of such compressed gradient methods heavily depend on $\omega$. For the family of unbiased compressors, $\omega$ can have linear dependence on $d$. For instance, $L_2$-quantization and unbiased RandK sparsifier (Beznosikov et al., 2023) achieves $\omega = \frac{d}{b} - 1$. While recent works (Szlendak et al., 2021) show that the convergence rate depends on the number of clients $C$, arguably in many practical settings with modern deep learning models, the number of parameters $d$ (hundreds of billions or more) is much larger than the number of active clients $C$ (millions). The usage of such unbiased compressors effectively leads to dimension-dependent convergence rate in compressed gradient based distributed learning methods such as MARINA (Gorbunov et al., 2021) and FedPAQ (Reisizadeh et al., 2020). (Song et al., 2023) is an inspiring work that introduces the sketch-and-desketch framework to distributed learning. However, the convergence bound derived in (Song et al., 2023) is limited since their communication cost scales linearly with the number of model parameters $d$, i.e., $O(d)$, which is also there in prior results using other compression algorithms.

**Biased Compression.** Biased gradient compressors achieve a lower variance in the compression error. TopK and biased RandK, which are commonly-used contractive compressors, achieve $\omega \leq (1 - \frac{b}{d})$. The issue of the biased methods leading to optimization divergence even in simple cases (Beznosikov et al., 2023) can be mitigated by introducing error feedback (EF) mechanisms (Seide et al., 2014). However, the state-of-the-art error feedback EF21 (Richtárik et al., 2021) still suffer from *distortion error* which is proportional to $\frac{d}{b}$. The dimensional dependence is inherited to the convergence rate of CocktailSGD (Wang et al., 2023) and 3PC (Richtárik et al., 2022) that employ biased gradient compressions. Further, most of the EF developments do not explicitly show compatibility with *adaptive methods, which involve anisotropic and nonlinear updates* (Tang et al., 2021).

**Communication.** The use of gradient compression calls for designing adequate transmission mechanisms. For sparsifying compressions, such as TopK and RandK, the average of sparse client gradients is possibly dense, which increases the downlink (server-to-client) transmission overhead. In the worst case, a plain average of the client gradients in MARINA (Gorbunov et al., 2021) leads to $bC$ in the number of non-zero bits. FetchSGD (Rothchild et al., 2020) mitigates the problem by adopting an extra call of topK compressor on the server side at additional compression costs.

**Additional Desiderata and Challenges.** Instead of an in-expectation analysis, which can hide high variance behavior, we want a high probability bound, which has to consider two inherent sources of noise in such settings: stochastic mini-batch and compression, as well as the third derived source of noise, due to the sequential dependencies of stochastic adaptive iterative updates. Further, while assuming gradient norms to be uniformly bounded simplifies analyses, we would like the guarantees to continue to hold without such an assumption. Finally, we want the analysis to hold when the local clients are doing multiple steps of updates, unlike the single update common in traditional distributed learning (Mishchenko et al., 2024; Tang et al., 2021; Wang et al., 2023).

# B Lemma for Random Sketching

For completeness, we provide the following lemmas that give high probability bounds on the inner products.

**Lemma B.1.** *(SRHT)[Same as Lemma D.23 Song et al. (2023)] Let $R \in \mathbb{R}^{b \times d}$ denote a subsample randomized Hadamard transform or AMS sketching matrix. Then for any fixed vector $h \in \mathbb{R}$ and any*

*fixed vector $g \in \mathbb{R}$ the following properties hold:*

$$\mathbb{P}\left[|\langle g^\top R^\top R h - g^\top h| \geq \frac{\log^{1.5}(d/\delta)}{\sqrt{b}}\|g\|_2\|h\|_2\right] \leq \Theta(\delta).$$

**Lemma B.2.** *(Gaussian)[Same as Lemma D.24 Song et al. (2023)] Let $R \in \mathbb{R}^{b \times d}$ denote a random Gaussian matrix. Then for any fixed vector $h \in \mathbb{R}$ and any fixed vector $g \in \mathbb{R}$ the following properties hold:*

$$\mathbb{P}\left[|\langle g^\top R^\top R h - g^\top h| \geq \frac{\log^{1.5}(d/\delta)}{\sqrt{b}}\|g\|_2\|h\|_2\right] \leq \Theta(\delta).$$

**Lemma B.3.** *(Count-Sketch)[Same as Lemma D.25 Song et al. (2023)] Let $R \in \mathbb{R}^{b \times d}$ denote a count-sketch matrix. Then for any fixed vector $h \in \mathbb{R}$ and any fixed vector $g \in \mathbb{R}$ the following properties hold:*

$$\mathbb{P}\left[|\langle g^\top R^\top R h - g^\top h| \geq \log(1/\delta)\|g\|_2\|h\|_2\right] \leq \Theta(\delta).$$

# C  Proof of Main Results

**Theorem C.1.** *Suppose the sequence of iterates $\{x_t\}_{t=1}^T$ is generated by Algorithm 1 (SADL) with a constant learning rate $\eta_t \equiv \eta$. Under Assumptions 1-4, for any $T$ and $\epsilon > 0$, with probability $1 - \Theta(\delta) - \mathcal{O}(\exp(-\Omega(\nu^2))) - \delta_g$,*

$$\kappa\eta J_1 K \sum_{t=1}^T \|\nabla\mathcal{L}(x_t)\|^2 \leq \mathcal{L}(z_1) + \frac{1}{\epsilon}\kappa\eta^2 LK^2 G^2 T + \nu\kappa\eta K\sqrt{T}\left(\frac{\log^{1.5}(CKTd/\delta)}{\sqrt{b}}\frac{G^2}{\epsilon} + \frac{\sigma}{\epsilon}\log^{\frac{1}{2}}\left(\frac{2T}{\delta_g}\right)\right)$$
$$+ \eta^2\kappa T\left(1 + \frac{\log^{1.5}(CKdT^2/\delta)}{\sqrt{b}}\right)^2 \frac{8\kappa\mathcal{I}LK^2 + 2G}{(1-\beta_1)^2}\frac{G^2}{\epsilon^2},$$

*where $\delta, \delta_g,$ and $\nu$ are the randomness of sketching, sub-Gaussian noise, and martingales respectively, and $J_1 := \left(\sqrt{1 + \frac{\log^{1.5}(CKd^2T^2/\delta)}{\sqrt{b}}}\eta KG + \epsilon\right)^{-1}.$*

**Remark 7.** *(Dependence on $K$)* The convergence bound in Theorem C.1 has a dependence on $K$. The primary focus of this work is to reduce the communication cost in FL algorithms, where the cost only depends on $T$ and sketching dimension $b$. Therefore, we view $K$ as a constant throughout the work. As we will show in Corollaries 1 and 2, if we set step-size $\eta$ as $\mathcal{O}(1/K\sqrt{T})$, which is the same as Reddi et al. (2020), the dependence on $K$ in the bound can be eliminated. $\square$

## C.1  Proof of Lemma C.2

Let

$$z_t = x_t + \frac{\beta_1}{1-\beta_1}(x_t - x_{t-1}) = \frac{1}{1-\beta_1}x_t - \frac{\beta_1}{1-\beta_1}x_{t-1}.$$

Then, the update on $z_t$ can be expressed as

$$
\begin{aligned}
z_{t+1} - z_t &= \frac{1}{1-\beta_1}(x_{t+1} - x_t) - \frac{\beta_1}{1-\beta_1}(x_t - x_{t-1}) \\
&= -\frac{1}{1-\beta_1}\kappa \hat{V}_t^{-1/2} \cdot m_t + \frac{\beta_1}{1-\beta_1}\kappa \hat{V_{t-1}}^{-1/2} \cdot m_{t-1} \\
&= -\frac{1}{1-\beta_1}\kappa \hat{V}_t^{-1/2} \cdot (\beta_1 m_{t-1} + (1-\beta_1)\cdot R_t^\top \bar{m}_t) + \frac{\beta_1}{1-\beta_1}\kappa \hat{V}_{t-1}^{-1/2}\cdot m_{t-1} \\
&= \frac{\beta_1}{1-\beta_1}\left(\kappa\hat{V}_{t-1}^{-1/2} - \kappa\hat{V}_t^{-1/2}\right)m_{t-1} - \frac{\kappa}{C}\hat{V}_t^{-1/2}R_t^\top \sum_{c=1}^{C}\bar{m}_t^c \\
&= \frac{\beta_1}{1-\beta_1}\left(\kappa\hat{V}_{t-1}^{-1/2} - \kappa\hat{V}_t^{-1/2}\right)m_{t-1} - \frac{\kappa}{C}\hat{V}_t^{-1/2}R_t^\top \sum_{c=1}^{C}R_t(x_{t,0}^c - x_{t,K}^c) \\
&= \frac{\beta_1}{1-\beta_1}\left(\kappa\hat{V}_{t-1}^{-1/2} - \kappa\hat{V}_t^{-1/2}\right)m_{t-1} - \frac{\kappa\eta}{C}\hat{V}_t^{-1/2}\sum_{c=1}^{C}\sum_{k=1}^{K}R_t^\top R_t g_{t,k}^c
\end{aligned}
$$

By Taylor expansion, we have

$$
\begin{aligned}
\mathcal{L}(z_{t+1}) &= \mathcal{L}(z_t) + \nabla\mathcal{L}(z_t)^\top(z_{t+1} - z_t) + \frac{1}{2}(z_{t+1} - z_t)^\top \hat{H}_{\mathcal{L}}(z_{t+1} - z_t) \\
&= \mathcal{L}(z_t) + \nabla\mathcal{L}(x_t)^\top(z_{t+1} - z_t) + (\nabla\mathcal{L}(z_t) - \nabla\mathcal{L}(x_t))^\top(z_{t+1} - z_t) + \frac{1}{2}(z_{t+1} - z_t)^\top \hat{H}_{\mathcal{L}}(z_{t+1} - z_t).
\end{aligned}
\tag{1}
$$

Bounding the first-order term

$$
\begin{aligned}
&\nabla\mathcal{L}(x_t)^\top(z_{t+1} - z_t) \\
=&\nabla\mathcal{L}(x_t)^\top\left(\frac{\beta_1}{1-\beta_1}\left(\kappa\hat{V}_{t-1}^{-1/2} - \kappa\hat{V}_t^{-1/2}\right)m_{t-1} - \frac{\kappa\eta}{C}\hat{V}_t^{-1/2}\sum_{c=1}^{C}\sum_{k=1}^{K}R_t^\top R_t g_{t,k}^c\right) \\
\leq&\frac{\beta_1}{1-\beta_1}\nabla\mathcal{L}(x_t)^\top\left(\kappa\hat{V}_{t-1}^{-1/2} - \kappa\hat{V}_t^{-1/2}\right)m_{t-1} - \frac{\eta}{C}\nabla\mathcal{L}(x_t)^\top(\kappa\hat{V}_t^{-1/2} - \kappa\hat{V}_{t-1}^{-1/2})\sum_{c=1}^{C}\sum_{k=1}^{K}R_t^\top R_t g_{t,k}^c \\
&- \frac{\kappa\eta}{C}\nabla\mathcal{L}(x_t)^\top\hat{V}_{t-1}^{-1/2}\sum_{c=1}^{C}\sum_{k=1}^{K}R_t^\top R_t g_{t,k}^c
\end{aligned}
$$

For the difference term, applying Lemma B.2 yields

$$
\begin{aligned}
&\frac{\eta}{C}\nabla\mathcal{L}(x_t)^\top(\kappa\hat{V}_t^{-1/2} - \kappa\hat{V}_{t-1}^{-1/2})\sum_{c=1}^{C}\sum_{k=1}^{K}R_t^\top R_t g_{t,k}^c \\
&\leq\frac{\eta\kappa}{C}(1 + \frac{\log^{1.5}(CKTd/\delta)}{\sqrt{b}})\|\nabla\mathcal{L}(x_t)\|\|\hat{V}_t^{-1/2} - \hat{V}_{t-1}^{-1/2}\|_2\sum_{c=1}^{C}\sum_{k=1}^{K}\|g_{t,k}^c\|
\end{aligned}
$$

Denote $[\cdot]_i$ as the $i$-th element of a vector. The $l2$-norm

$$
\begin{aligned}
\|\hat{V}_t^{-1/2} - \hat{V}_{t-1}^{-1/2}\|_2 &= \max_i \frac{1}{\sqrt{\hat{v}_{t-1,i}} + \epsilon} - \frac{1}{\sqrt{\hat{v}_{t,i}} + \epsilon} = \max_i \frac{\sqrt{\hat{v}_{t,i}} - \sqrt{\hat{v}_{t-1,i}}}{(\sqrt{\hat{v}_{t-1,i}} + \epsilon)(\sqrt{\hat{v}_{t,i}} + \epsilon)} \\
&= \max_i \frac{\hat{v}_{t,i} - \hat{v}_{t-1,i}}{(\sqrt{\hat{v}_{t-1,i}} + \epsilon)(\sqrt{\hat{v}_{t,i}} + \epsilon)(\sqrt{\hat{v}_{t,i}} + \sqrt{\hat{v}_{t-1,i}})}
\end{aligned}
$$

By definition, $\hat{v}_t = \max(\hat{v}_{t-1}, v_t)$. If $\hat{v}_{t,i} = \hat{v}_{t-1,i}$, the RHS is 0. Otherwise, $\hat{v}_{t,i} = v_{t,i}$.

$$
\begin{aligned}
\|\hat{V}_t^{-1/2} - \hat{V}_{t-1}^{-1/2}\|_2 &\leq \max_i \frac{v_{t,i} - v_{t-1,i}}{(\sqrt{\hat{v}_{t-1,i}} + \epsilon)(\sqrt{\hat{v}_{t,i}} + \epsilon)(\sqrt{\hat{v}_{t,i}} + \sqrt{\hat{v}_{t-1,i}})} \\
&\leq \max_i \frac{(1-\beta_2)(\bar{v}_{t,i} - v_{t-1,i})}{\epsilon^2 \sqrt{(1-\beta_2)\bar{v}_{t,i}}} \\
&\leq \max_i \frac{\sqrt{1-\beta_2}}{\epsilon^2}\sqrt{\bar{v}_{t,i}} \\
&= \frac{\sqrt{1-\beta_2}}{\epsilon^2} \max_i \sqrt{\frac{\eta^2}{C}\sum_{c=1}^{C}\sum_{k=1}^{K}[(\sum R_t^\top R_t g_{t,k}^c)^2]_i} \\
&\leq \frac{2\eta\sqrt{(1-\beta_2)}}{\epsilon^2}\sqrt{1 + \frac{\log^{1.5}(CKtd^2/\delta)}{\sqrt{b}}}G.
\end{aligned}
$$

The first inequality is from $\hat{v}_{t-1,i} \geq v_{t-1,i}$. The second inequality comes from $\hat{v}_{t,i} \geq v_{t,i} \geq (1-\beta_2)\bar{v}_{t,i}$. The last inequality follows from applying Lemma B.2 to each dimension of $g_{t,k}^c$. Plugging into the bound for the difference term

$$
\begin{aligned}
&\frac{\eta}{C}\nabla\mathcal{L}(x_t)^\top(\kappa\hat{V}_t^{-1/2} - \kappa\hat{V}_{t-1}^{-1/2})\sum_{c=1}^{C}\sum_{k=1}^{K}R_t^\top R_t g_{t,k}^c \\
&\leq \frac{2\eta^2\kappa\sqrt{(1-\beta_2)}}{\epsilon^2}(1 + \frac{\log^{1.5}(CKtd^2/\delta)}{\sqrt{b}})^{3/2}G^3
\end{aligned}
$$

The quadratic terms can be written as

$$
(\nabla\mathcal{L}(z_t) - \nabla\mathcal{L}(x_t))^\top(z_{t+1} - z_t) = (z_t - x_t)^\top\hat{H}_{\mathcal{L}}(\frac{1}{1-\beta_1}(x_{t+1} - x_t) - \frac{\beta_1}{1-\beta_1}(x_t - x_{t-1})),
$$

where $\hat{H}_{\mathcal{L}}$ is a second-order Taylor remainder. So the quadratic term can be further seen as a quadratic form over $z_{t+1} - z_t$ and $z_t - x_t$, denote as $\mathcal{Q}(z_{t+1} - z_t, z_t - x_t)$. For the same reason, the term $\frac{1}{2}(z_{t+1} - z_t)^\top\hat{H}_{\mathcal{L}}(z_{t+1} - z_t)$ can also be written into a quadratic form $\mathcal{Q}(z_{t+1} - z_t, z_{t+1} - z_t)$. Putting the two terms together yields a quadratic form of $\mathcal{Q}(z_{t+1} - z_t, z_t - x_t)$.

Overall, the descent lemma can be written as

$$
\begin{aligned}
&\mathcal{L}(z_{t+1}) \\
&\leq \mathcal{L}(z_t) + \frac{2\eta^2\kappa\sqrt{1-\beta_2}}{(1-\beta_1)\epsilon^2}(1 + \frac{\log^{1.5}(CKtd^2/\delta)}{\sqrt{b}})^{3/2}G^3 - \frac{\kappa\eta}{C}\nabla\mathcal{L}(x_t)^\top\hat{V}_{t-1}^{-1/2}\sum_{c=1}^{C}\sum_{k=1}^{K}R_t^\top R_t g_{t,k}^c \\
&\quad + (\nabla\mathcal{L}(z_t) - \nabla\mathcal{L}(x_t))^\top(z_{t+1} - z_t) + \frac{1}{2}(z_{t+1} - z_t)^\top\hat{H}_{\mathcal{L}}(z_{t+1} - z_t).
\end{aligned}
$$

## C.2    Proof of Lemma C.3 (Generalized version of Lemma 2.4)

*Proof.* We can prove by induction. For $t = 0$, since $m_0 = 0$, the inequality holds. Suppose we have for $h \in \mathbb{R}^d$, s.t. $\|h\| \leq H$, with probability $1 - \Theta((t-1)\delta)$,

$$
|m_{t-1}^\top h| \leq (1 + \frac{\log^{1.5}(CKd/\delta)}{\sqrt{b}})G
$$

Then by the update rule,

$$|m_t^\top h| = |(\beta_1 \cdot m_{t-1} + (1 - \beta_1) \cdot \frac{\eta}{C} \sum_{c=1}^{C} \sum_{k=1}^{K} R_t^\top R_t g_{t,k}^c)^\top h|$$

$$\leq \beta_1 |m_{t-1}^\top h| + \frac{(1-\beta_1)\eta}{C} \sum_{c=1}^{C} \sum_{k=1}^{K} |\langle R_t^\top R_t g_{t,k}^c, h \rangle|$$

$$\leq \beta_1 |m_{t-1}^\top h| + (1-\beta_1)(1 + \frac{\log^{1.5}(CKd/\delta)}{\sqrt{b}})\eta \sum_{k=1}^{K} \|g_{t,k}^c\|_2 \|h\|_2$$

$$\leq (1 + \frac{\log^{1.5}(CKd/\delta)}{\sqrt{b}})\eta KGH, \quad w.p. \ 1 - \Theta(t\delta).$$

Let $h = \hat{V}_t^{-1/2} v_i$. Then $\|h\|_2 \leq 1/\epsilon$. We have

$$|(\hat{V}_t^{-1/2} m_t)^\top v_i| \leq (1 + \frac{\log^{1.5}(CKd/\delta)}{\sqrt{b}})\eta KG/\epsilon$$

$\square$

## C.3   Proof of Lemma 2.5

We first prove the element-wise lower bound of the diagonal matrix $\hat{V}_{t-1}^{-1/2}$. Denote $(\hat{V}_{t-1}^{-1/2})_i$ as the $i$-th element on the diagonal of $\hat{V}_{t-1}^{-1/2}$. By the update rule,

$$(\hat{V}_{t-1}^{-1/2})_i \geq (\max_{t-1}(\sqrt{v_{t,i}}) + \epsilon)^{-1} \geq (\sqrt{1 + \frac{\log^{1.5}(CKtd/\delta)}{\sqrt{b}}}\eta KG + \epsilon)^{-1}, \quad w.p. \ 1 - \Theta(\delta)$$

where the last inequality follows by letting $h$ as a one-hot vector $h_i$ in Lemma B.1, observing that the elements can be transformed to an inner product form $v_{t,i} = v_t^\top h_i$. Then the scaled gradient norm can be lower bounded as

$$\nabla\mathcal{L}(x_t)^\top \hat{V}_{t-1}^{-1/2} \nabla\mathcal{L}(x_t) \geq \min_i(\hat{V}_{t-1}^{-1/2})_i \sum_{i=1}^{d} [\nabla\mathcal{L}(x_t)]_i^2$$

$$\geq (\sqrt{1 + \frac{\log^{1.5}(CKtd/\delta)}{\sqrt{b}}}\eta KG + \epsilon)^{-1}\|\nabla\mathcal{L}(x_t)\|^2, \quad w.p. \ 1 - \Theta(d\delta)$$

which completes the proof by applying union bounded on the dimension $d$.

## C.4   Proof of Lemma 2.6

Since the noise is zero-centered, we view the random process of

$$\{Y_t = \sum_{\tau=1}^{t} \frac{1}{C} \sum_{c=1}^{C} \sum_{k=1}^{K} \nabla\mathcal{L}(x_\tau)^\top \hat{V}_{\tau-1}^{-1/2}(R_\tau^\top R_\tau g_{\tau,k}^c - g_{\tau,k}^c)\}_{t=1}^{T}$$

as a martingale. The difference of $|Y_{t+1} - Y_t|$ is bounded with high probability

$$|Y_{t+1} - Y_t| = \sum_{k=1}^{K} |\nabla\mathcal{L}(x_t)^\top \hat{V}_{t-1}^{-1/2}(R_t^\top R_t g_{t,k}^c - g_{t,k}^c)| \leq \frac{\log^{1.5}(CKd/\delta)}{\sqrt{b}} KG\|\hat{V}_t^{-1/2}\nabla\mathcal{L}(x_t)\|_2,$$

$$w.p. \ 1 - \Theta(\delta)$$

Then by Azuma's inequality,

$$\mathbb{P}(|Y_T| \geq \nu \sqrt{\sum_{t=1}^{T} \left(\frac{\log^{1.5}(CKd/\delta)}{\sqrt{b}} KG\|\hat{V}_t^{-1/2}\nabla\mathcal{L}(x_t)\|_2\right)^2}) = O(\exp(-\Omega(\nu^2))) + T\delta$$

Note that the original Azuma's is conditioned on a uniform bound of the difference term, while our bound here is of high probability. Hence, we need another union bound. A similar bound can be achieved for the sub-Gaussian noise in stochastic gradient. Let

$$Z_t = \sum_{\tau=1}^{t} \frac{1}{C} \sum_{c=1}^{C} \sum_{k=1}^{K} \nabla\mathcal{L}(x_\tau)^\top \hat{V}_{\tau-1}^{-1/2}(g_{\tau,k}^c - \nabla\mathcal{L}^c(x_{t,k}^c)).$$

Then

$$\mathbb{P}(|Z_T| \geq \nu K \sqrt{\sum_{t=1}^{T} \frac{\sigma^2}{\epsilon^2} \log(\frac{2T}{\delta_g})}) = O(\exp(-\Omega(\nu^2))) + \delta_g$$

Combining the two bounds by union bound completes the proof.

### C.5 Proof of Theorem C.1

We first introduce the lemma

**Lemma C.2.** *For any round $t \in [T]$,*

$$\mathcal{L}(z_{t+1}) \leq \mathcal{L}(z_t) - \frac{\kappa\eta}{C} \sum_{c=1}^{C} \sum_{k=1}^{K} \nabla\mathcal{L}(x_t)^\top \hat{V}_{t-1}^{-1/2} R_t^\top R_t g_{t,k}^c + (z_{t+1} - z_t)^\top H_\mathcal{L}(\hat{z}_t)(z_{t+1} - z_t)$$

$$+ \frac{2\eta^2\kappa}{(1-\beta_1)\epsilon^2} + (\nabla\mathcal{L}(z_t) - \nabla\mathcal{L}(x_t))^\top (z_{t+1} - z_t) + (1 + \frac{\log^{1.5}(CKtd^2/\delta)}{\sqrt{b}})^{3/2} G^3,$$

*where $H_\mathcal{L}(\hat{z}_t)$ is the loss Hessian at some $\hat{z}_t$ within the element-wise interval of $[x_t, z_t]$*

After applying Lemma C.2. The second order quadratic forms in the descent lemma can be written as

$$(\nabla\mathcal{L}(z_t) - \nabla\mathcal{L}(x_t))^\top (z_{t+1} - z_t)$$
$$= (z_t - x_t)^\top \hat{H}_\mathcal{L}(\frac{1}{1-\beta_1}(x_{t+1} - x_t) - \frac{\beta_1}{1-\beta_1}(x_t - x_{t-1}))$$
$$= -\kappa\frac{\beta_1}{1-\beta_1}(\hat{V}_{t-1}^{-1/2}m_{t-1})^\top \hat{H}_\mathcal{L}(\frac{1}{1-\beta_1}(-\kappa\hat{V}_t^{-1/2}m_t) - \frac{\beta_1}{1-\beta_1}(-\kappa\hat{V}_{t-1}^{-1/2}m_{t-1}))$$
$$= \kappa^2 \frac{\beta_1}{(1-\beta_1)^2}(\hat{V}_{t-1}^{-1/2}m_{t-1})^\top \hat{H}_\mathcal{L}(\hat{V}_t^{-1/2}m_t) - \kappa^2 \frac{\beta_1^2}{(1-\beta_1)^2}(\hat{V}_{t-1}^{-1/2}m_{t-1})^\top \hat{H}_\mathcal{L}(\hat{V}_{t-1}^{-1/2}m_{t-1}),$$

and

$$(z_{t+1} - z_t)^\top \hat{H}_\mathcal{L}(z_{t+1} - z_t)$$
$$= (\frac{1}{1-\beta_1}(x_{t+1} - x_t) - \frac{\beta_1}{1-\beta_1}(x_t - x_{t-1}))^\top \hat{H}_\mathcal{L}(\frac{1}{1-\beta_1}(x_{t+1} - x_t) - \frac{\beta_1}{1-\beta_1}(x_t - x_{t-1}))$$
$$= \frac{1}{(1-\beta_1)^2}(x_{t+1} - x_t)^\top \hat{H}_\mathcal{L}(x_{t+1} - x_t) - \frac{2\beta_1}{(1-\beta_1)^2}(x_{t+1} - x_t)^\top \hat{H}_\mathcal{L}(x_t - x_{t-1})$$
$$+ \frac{\beta_1^2}{(1-\beta_1)^2}(x_t - x_{t-1})^\top \hat{H}_\mathcal{L}(x_t - x_{t-1}), \tag{2}$$

which is essentially a quadratic form defined on $\hat{V}_t^{-1/2}m_t$ and $\hat{V}_{t-1}^{-1/2}m_{t-1}$. Hence, we provide a generalized version of Lemma 2.4, as follows.

**Lemma C.3.** *With probability $1-\Theta(t\delta)$, for eigenvector $v_i$ of the Hessian matrix, $|(\hat{V}_t^{-1/2}m_t)^\top v_i| \leq (1 + \frac{\log^{1.5}(CKd/\delta)}{\sqrt{b}})\eta KG/\epsilon$.*

Note that $v_i$ can be any basis and is constant throughout the training process. Then the sum of quadratic forms is written as

$$
(\nabla\mathcal{L}(z_t) - \nabla\mathcal{L}(x_t))^\top (z_{t+1} - z_t)
$$

$$
\leq \kappa^2 \frac{\beta_1}{(1-\beta_1)^2}(\hat{V}_{t-1}^{-1/2}m_{t-1})^\top \hat{H}_{\mathcal{L}}(\hat{V}_t^{-1/2}m_t) - \kappa^2 \frac{\beta_1^2}{(1-\beta_1)^2}(\hat{V}_{t-1}^{-1/2}m_{t-1})^\top \hat{H}_{\mathcal{L}}(\hat{V}_{t-1}^{-1/2}m_{t-1}),
$$

$$
= \kappa^2 \frac{\beta_1}{(1-\beta_1)^2}\sum_{i=1}^d \lambda_i(\hat{V}_{t-1}^{-1/2}m_{t-1})^\top (v_iv_i^\top)\hat{V}_t^{-1/2}m_t - \kappa^2 \frac{\beta_1^2}{(1-\beta_1)^2}\sum_{i=1}^d \lambda_i(\hat{V}_{t-1}^{-1/2}m_{t-1})^\top (v_iv_i^\top)\hat{V}_{t-1}^{-1/2}m_{t-1}
$$

$$
\leq \kappa^2 \frac{\beta_1}{(1-\beta_1)^2}\sum_{i=1}^d |\lambda_i|\,|(\hat{V}_{t-1}^{-1/2}m_{t-1})^\top v_i|\,|(\hat{V}_t^{-1/2}m_t)^\top v_i| + \kappa^2 \frac{\beta_1^2}{(1-\beta_1)^2}\sum_{i=1}^d |\lambda_i|\,|(\hat{V}_{t-1}^{-1/2}m_{t-1})^\top v_i|^2
$$

$$
\leq \kappa^2 \frac{2}{(1-\beta_1)^2}\mathcal{I}L(1 + \frac{\log^{1.5}(CKd/\delta)}{\sqrt{b}})^2\eta^2 K^2 G^2/\epsilon^2, \tag{3}
$$

where the last inequality is by $\beta_1 \leq 1$ and Lemma. C.3.

**First-Order Descent Term**. The first-order term in the descent lemma can be decomposed into three components, which we will handle separately.

$$
\nabla\mathcal{L}(x_t)^\top \hat{V}_{t-1}^{-1/2}R_t^\top R_t g_{t,k}^c = \underbrace{\nabla\mathcal{L}(x_t)^\top \hat{V}_{t-1}^{-1/2}\nabla\mathcal{L}^c(x_t)}_{\mathcal{D}_1} + \underbrace{\nabla\mathcal{L}(x_t)^\top \hat{V}_{t-1}^{-1/2}(R_t^\top R_t g_{t,k}^c - \nabla\mathcal{L}^c(x_{t,k}^c))}_{\mathcal{D}_2}
$$

$$
+ \underbrace{\nabla\mathcal{L}(x_t)^\top \hat{V}_{t-1}^{-1/2}(\nabla\mathcal{L}^c(x_{t,k}^c) - \nabla\mathcal{L}^c(x_t))}_{\mathcal{D}_3}.
$$

First, $\mathcal{D}_3$ can be reduced to a second-order term by smoothness over $\mathcal{L}$,

$$
\nabla\mathcal{L}(x_t)^\top \hat{V}_{t-1}^{-1/2}(\nabla\mathcal{L}^c(x_{t,k}^c) - \nabla\mathcal{L}^c(x_t)) = \nabla\mathcal{L}(x_t)^\top \hat{V}_{t-1}^{-1/2}\hat{H}_L^c(x_{t,k}^c - x_t)
$$

$$
= -\eta \sum_{\tau=1}^k \nabla\mathcal{L}(x_t)^\top \hat{V}_{t-1}^{-1/2}\hat{H}_{\mathcal{L}}^c g_{t,\tau}^c
$$

$$
\geq -\frac{\eta}{\epsilon}L\|\nabla\mathcal{L}\|\sum_{\tau=1}^k \|g_{t,\tau}^c\| \geq -\frac{1}{\epsilon}\eta LKG^2. \tag{4}
$$

Note that this term does not involve any stochasticity with regard to random sketching, which means we can directly derive the upper bound by Cauchy-Schwartz in the last inequality.

Next observing that $\frac{1}{C}\sum_{c=1}^C \nabla\mathcal{L}^c(x_t) = \nabla\mathcal{L}(x_t)$, $\mathcal{D}_1$ composes a scaled squared gradient norm. Applying element-wise high probability bound on random sketching yields the lower bound for the scale. By Lemma 2.5, we can derive the lower bound for $\mathcal{D}_1$. Note that applying union bound to $\mathcal{D}_1$ does not introduce another $T$ dependence, since $\hat{v}_{t,i}$ is monotonically non-decreasing.

**Martingale for zero-centered noise.** $\mathcal{D}_2$ contains a zero-centered noise term $R_t^\top R_t g_{t,k}^c - \nabla\mathcal{L}^c(x_{t,k}^c)$, where the randomness is over $R_t$ and the mini-batch noise at round $t$. Despite $x_{t,k}^c$ has temporal dependence, the fresh noise at round $t$ is independent of the randomness in the previous iterations. Hence, the random process defined by the aggregation of these norm terms over time forms a martingale. By Lemma 2.6, we can bound this term $\mathcal{D}_2$.

Finally, putting these parts together by union bound over $[T]$ and telescoping the descent lemma leads to Theorem C.1. To show this, with probability $1 - \Theta(\delta)$,

$$\mathcal{L}(z_{t+1})$$

$$\overset{(i)}{\leq} \mathcal{L}(z_t) - \frac{\kappa\eta}{C}\sum_{c=1}^{C}\sum_{k=1}^{K}\nabla\mathcal{L}(x_t)^\top \hat{V}_{t-1}^{-1/2}R_t^\top R_t g_{t,k}^c + (z_{t+1} - z_t)^\top H_\mathcal{L}(\hat{z}_t)(z_{t+1} - z_t)$$

$$+ \frac{2\eta^2\kappa}{(1-\beta_1)\epsilon^2}(1 + \frac{\log^{1.5}(CKtd^2/\delta)}{\sqrt{b}})^{3/2}G^3 + (\nabla\mathcal{L}(z_t) - \nabla\mathcal{L}(x_t))^\top(z_{t+1} - z_t)$$

$$\overset{(ii)}{\leq} \mathcal{L}(z_t) - \kappa\eta(\sqrt{1 + \frac{\log^{1.5}(CKtd^2/\delta)}{\sqrt{b}}}\eta KG + \epsilon)^{-1}\|\nabla\mathcal{L}(x_t)\|^2 + \frac{1}{\epsilon}\kappa\eta^2 LKG^2$$

$$+ \frac{\kappa\eta}{C}\sum_{c=1}^{C}\sum_{k=1}^{K}\nabla\mathcal{L}(x_t)^\top \hat{V}_{t-1}^{-1/2}(R_t^\top R_t g_{t,k}^c - \nabla\mathcal{L}^c(x_{t,k}^c))$$

$$+ \kappa^2\frac{2}{(1-\beta_1)^2}\mathcal{I}L(1 + \frac{\log^{1.5}(CKd/\delta)}{\sqrt{b}})^2\eta^2 K^2 G^2/\epsilon^2$$

$$+ \frac{2\eta^2\kappa}{(1-\beta_1)\epsilon^2}(1 + \frac{\log^{1.5}(CKtd^2/\delta)}{\sqrt{b}})^{3/2}G^3 + (z_{t+1} - z_t)^\top H_\mathcal{L}(\hat{z}_t)(z_{t+1} - z_t)$$

$$\overset{(iii)}{\leq} \mathcal{L}(z_t) - \kappa\eta(\sqrt{1 + \frac{\log^{1.5}(CKtd^2/\delta)}{\sqrt{b}}}\eta KG + \epsilon)^{-1}\|\nabla\mathcal{L}(x_t)\|^2 + \frac{1}{\epsilon}\kappa\eta^2 LKG^2$$

$$+ \frac{\kappa\eta}{C}\sum_{c=1}^{C}\sum_{k=1}^{K}\nabla\mathcal{L}(x_t)^\top \hat{V}_{t-1}^{-1/2}(R_t^\top R_t g_{t,k}^c - \nabla\mathcal{L}^c(x_{t,k}^c))$$

$$+ \kappa^2\frac{2+(1+\beta_1)^2}{(1-\beta_1)^2}\mathcal{I}L(1 + \frac{\log^{1.5}(CKd/\delta)}{\sqrt{b}})^2\eta^2 K^2 G^2/\epsilon^2$$

$$+ \frac{2\eta^2\kappa}{(1-\beta_1)\epsilon^2}(1 + \frac{\log^{1.5}(CKtd^2/\delta)}{\sqrt{b}})^{3/2}G^3$$

where $(i)$ follows from Lemma. C.2. $(ii)$ follows from (3), (4) and Lemma 2.5. $(iii)$ follows from (2).

Summing up with $t \in [T]$ and moving the squared gradient norm term to the left hand side yields that with probability $1 - \Theta(\delta) - O(\exp(-\Omega(\nu^2))) - \delta_g$,

$$\kappa\eta(\sqrt{1 + \frac{\log^{1.5}(CKd^2T^2/\delta)}{\sqrt{b}}}\eta KG + \epsilon)^{-1}\sum_{t=1}^{T}\|\nabla\mathcal{L}(x_t)\|^2$$

$$\leq \mathcal{L}(z_1) + \frac{1}{\epsilon}\kappa\eta^2 LKG^2 T + \kappa^2\eta^2 T\frac{2+(1+\beta_1)^2}{(1-\beta_1)^2}\mathcal{I}L(1 + \frac{\log^{1.5}(CKd/\delta)}{\sqrt{b}})^2 K^2 G^2/\epsilon^2$$

$$+ \frac{2\eta^2\kappa T}{(1-\beta_1)\epsilon^2}(1 + \frac{\log^{1.5}(CKtd^2/\delta)}{\sqrt{b}})^{3/2}G^3$$

$$+ \sum_{t=1}^{T}\frac{\kappa\eta}{C}\sum_{c=1}^{C}\sum_{k=1}^{K}\nabla\mathcal{L}(x_t)^\top \hat{V}_{t-1}^{-1/2}(R_t^\top R_t g_{t,k}^c - \nabla\mathcal{L}^c(x_{t,k}^c))$$

$$\overset{(i)}{\leq} \mathcal{L}(z_1) + \frac{1}{\epsilon}\kappa\eta^2 LKG^2 T + \kappa^2\eta^2 T\frac{2+(1+\beta_1)^2}{(1-\beta_1)^2}\mathcal{I}L(1 + \frac{\log^{1.5}(CKd/\delta)}{\sqrt{b}})^2 K^2 G^2/\epsilon^2$$

$$+ \frac{2\eta^2\kappa T}{(1-\beta_1)\epsilon^2}(1 + \frac{\log^{1.5}(CKtd^2/\delta)}{\sqrt{b}})^{3/2}G^3$$

$$+ \nu\kappa\eta K\sqrt{T}(\frac{\log^{1.5}(CKTd/\delta)}{\sqrt{b}}\frac{G^2}{\epsilon} + \frac{\sigma}{\epsilon}\log^{\frac{1}{2}}(\frac{2T}{\delta_g})),$$

where $(i)$ follows from Lemma. 2.6. We conclude the proof by realizing $\beta_1 \leq 1$.

## C.6 Proof of Corollary 1

In the aysmptotic regime, with sufficiently large $T$, the term $\sqrt{1 + \frac{\log^{1.5}(CKd^2T^2/\delta)}{\sqrt{b}}}\eta KG$ approaches $\epsilon$, so the denominator on the LHS can be replaced with $2\epsilon$. Then the derivation is straightforward by just substituting $\eta = \frac{1}{\sqrt{T}K}$ into Theorem C.1.

## C.7 Proof of Corollary 2

We first develop the convergence bound in Theorem C.1 under the condition $b \geq \log^3(CKd^2T^2/\delta)$,

$$\left(\sqrt{2}\eta KG + \epsilon\right)^{-1} \kappa\eta K \sum_{t=1}^{T} \|\nabla\mathcal{L}(x_t)\|^2 \leq \mathcal{L}(z_1) + \frac{1}{\epsilon}\kappa\eta^2 LK^2G^2T$$

$$+ \nu\kappa\eta K\sqrt{T}(\frac{G^2}{\epsilon} + \frac{\sigma}{\epsilon}\log^{\frac{1}{2}}(\frac{2T}{\delta_g})) + \eta^2\kappa^2T\frac{32}{(1-\beta_1)^2}\frac{\mathcal{I}LK^2G^2}{\epsilon^2},$$

The condition on $T \leq \frac{J_3 - \sqrt{2}G}{\epsilon^2}$ is equivalent to

$$\frac{\sqrt{2}\eta KG + \epsilon}{\eta K} \leq J_3,$$

since $\eta = \frac{1}{\sqrt{T}K}$. Then scaling the coefficient on the left hand side and substituting $\frac{1}{\sqrt{T}K}$ for $\eta$, we derive

$$\frac{1}{J_3T} \sum_{t=1}^{T} \|\nabla\mathcal{L}(x_t)\|^2 \leq \frac{\mathcal{L}(z_1)\epsilon}{\kappa T} + \frac{1}{\epsilon}\frac{LG^2}{T} + \frac{\nu}{T}(G^2 + \sigma\log^{\frac{1}{2}}(\frac{2T}{\delta_g})) + \frac{\kappa}{T}\frac{32}{(1-\beta_1)^2}\frac{\mathcal{I}LG^2}{\epsilon},$$

## C.8 A non-asymptotic bound on practical learning rates

We first state a convergence bound on using practical learning rates, which decays as the optimization procedure.

**Theorem C.4.** *Suppose the sequence of iterates $\{x_t\}_{t=1}^{T}$ is generated by Algorithm 1 with a decaying learning rate $\eta_t = \frac{1}{\sqrt{t+T_0}K}$, where $T_0 = \lceil\frac{1}{1-\beta_1^2}\rceil$. Under Assumptions 1-4, for any $T$ and $\epsilon > 0$, with probability $1 - \Theta(\delta) - O(\exp(-\Omega(\nu^2))) - \delta_g$,*

$$\sum_{t=1}^{T} \left(\sqrt{1 + \frac{\log^{1.5}(CKd^2T^2/\delta)}{\sqrt{b}}}\eta_t JKG + \epsilon\right)^{-1} \kappa\eta_t\|\nabla\mathcal{L}(x_t)\|^2 \leq \mathcal{L}(z_1) + \frac{1}{\epsilon}\kappa LG^2\log T$$

$$+ \nu\kappa\log T(\frac{\log^{1.5}(CKTd/\delta)}{\sqrt{b}}\frac{G^2}{\epsilon} + \frac{\sigma}{\epsilon}\log^{\frac{1}{2}}(\frac{2T}{\delta_g})) + \kappa^2\log T(1 + \frac{\log^{1.5}(CKdT^2/\delta)}{\sqrt{b}})^2\frac{8}{(1-\beta_1)^2}\frac{\mathcal{I}LG^2}{\epsilon^2},$$

*where $\delta$, $\delta_g$, and $\nu$ are the randomness from sketching, sub-Gaussian stochastic noise and martingales respectively, and $J$ is a constant defined in Lemma. C.5.*

Alike the analysis in the constant learning rate case, we first define auxiliary variables $z_t$

$$z_t = x_t + \frac{\beta_1}{1-\beta_1}(x_t - x_{t-1}) = \frac{1}{1-\beta_1}x_t - \frac{\beta_1}{1-\beta_1}x_{t-1}.$$

Then, the update on $z_t$ can be expressed as

$$z_{t+1} - z_t = \frac{1}{1-\beta_1}(x_{t+1} - x_t) - \frac{\beta_1}{1-\beta_1}(x_t - x_{t-1})$$

$$= \frac{\beta_1}{1-\beta_1}\left(\kappa\hat{V}_{t-1}^{-1/2} - \kappa\hat{V}_t^{-1/2}\right)m_{t-1} - \frac{\kappa\eta_t}{C}\hat{V}_t^{-1/2}\sum_{c=1}^{C}\sum_{k=1}^{K}R_t^\top R_t g_{t,k}^c$$

By Taylor expansion, we have

$$\mathcal{L}(z_{t+1}) = \mathcal{L}(z_t) + \nabla\mathcal{L}(z_t)^\top(z_{t+1} - z_t) + \frac{1}{2}(z_{t+1} - z_t)^\top \hat{H}_\mathcal{L}(z_{t+1} - z_t)$$

$$= \mathcal{L}(z_t) + \nabla\mathcal{L}(x_t)^\top(z_{t+1} - z_t) + (\nabla\mathcal{L}(z_t) - \nabla\mathcal{L}(x_t))^\top(z_{t+1} - z_t) + \frac{1}{2}(z_{t+1} - z_t)^\top \hat{H}_\mathcal{L}(z_{t+1} - z_t).$$

Bounding the first-order term

$$\nabla\mathcal{L}(x_t)^\top(z_{t+1} - z_t)$$

$$= \nabla\mathcal{L}(x_t)^\top \left( \frac{\beta_1}{1-\beta_1}\left(\kappa\hat{V}_{t-1}^{-1/2} - \kappa\hat{V}_t^{-1/2}\right)m_{t-1} - \frac{\kappa\eta_t}{C}\hat{V}_t^{-1/2}\sum_{c=1}^C\sum_{k=1}^K R_t^\top R_t g_{t,k}^c \right)$$

$$\leq \frac{\beta_1}{1-\beta_1}\|\nabla\mathcal{L}(x_t)\|_\infty(\|\kappa\hat{V}_{t-1}^{-1/2}\|_{1,1} - \|\kappa\hat{V}_t^{-1/2}\|_{1,1})\|m_{t-1}\|_\infty$$

$$- \frac{\eta_t}{C}\nabla\mathcal{L}(x_t)^\top(\kappa\hat{V}_t^{-1/2} - \kappa\hat{V}_{t-1}^{-1/2})\sum_{c=1}^C\sum_{k=1}^K R_t^\top R_t g_{t,k}^c - \frac{\kappa\eta_t}{C}\nabla\mathcal{L}(x_t)^\top\hat{V}_{t-1}^{-1/2}\sum_{c=1}^C\sum_{k=1}^K R_t^\top R_t g_{t,k}^c$$

$$\leq \left( \frac{\beta_1}{1-\beta_1}\|m_{t-1}\|_\infty + \frac{\eta_t}{C}\|\sum_{c=1}^C\sum_{k=1}^K R_t^\top R_t g_{t,k}^c\|_\infty \right)\|\nabla\mathcal{L}(x_t)\|_\infty(\|\kappa\hat{V}_{t-1}^{-1/2}\|_{1,1} - \|\kappa\hat{V}_t^{-1/2}\|_{1,1})$$

$$- \frac{\kappa\eta_t}{C}\sum_{c=1}^C\sum_{k=1}^K \nabla\mathcal{L}(x_t)^\top\hat{V}_{t-1}^{-1/2}R_t^\top R_t g_{t,k}^c.$$

The quadratic terms can be written as

$$(\nabla\mathcal{L}(z_t) - \nabla\mathcal{L}(x_t))^\top(z_{t+1} - z_t) = (z_t - x_t)^\top \hat{H}_\mathcal{L}\left(\frac{1}{1-\beta_1}(x_{t+1} - x_t) - \frac{\beta_1}{1-\beta_1}(x_t - x_{t-1})\right),$$

where $\hat{H}_\mathcal{L}$ is a second-order Taylor remainder.

To bound the quadratic term, the counterpart of Lemma C.3 can be stated as

**Lemma C.5.** *With learning rate $\eta_t = O(\frac{1}{\sqrt{t+T_0}})$, where $T_0 = \lceil\frac{1}{1-\beta_1^2}\rceil$. Denote $J = \frac{1-\beta_1}{\sqrt{T_0+1}}/(\frac{1}{\sqrt{T_0+1}} - \frac{\beta_1}{\sqrt{T_0}})$. Then with probability $1 - \Theta(t\delta)$,*

$$|m_{t-1}^\top h| \leq (1 + \frac{\log^{1.5}(CKd/\delta)}{\sqrt{b}})JKGH$$

*Proof.* For $t = 0$, since $m_0 = 0$, the inequality holds. Suppose we have for $h \in \mathbb{R}^d$, s.t. $\|h\| \leq H$, with probability $1 - \Theta((t-1)\delta)$,

$$|m_{t-1}^\top h| \leq (1 + \frac{\log^{1.5}(CKd/\delta)}{\sqrt{b}})JKGH$$

By the update rule,

$$|m_t^\top h| = |(\beta_1 \cdot m_{t-1} + (1 - \beta_1) \cdot \frac{\eta}{C}\sum_{c=1}^C\sum_{k=1}^K R_t^\top R_t g_{t,k}^c)^\top h|$$

$$\leq \beta_1|m_{t-1}^\top h| + \frac{(1-\beta_1)\eta}{C}\sum_{c=1}^C\sum_{k=1}^K |\langle R_t^\top R_t g_{t,k}^c, h\rangle|$$

$$\leq \beta_1|m_{t-1}^\top h| + (1 - \beta_1)(1 + \frac{\log^{1.5}(CKd/\delta)}{\sqrt{b}})\eta_t\sum_{k=1}^K \|g_{t,k}^c\|_2\|h\|_2$$

$$\leq (1 + \frac{\log^{1.5}(CKd/\delta)}{\sqrt{b}})\eta_t JKGH, \quad w.p.\ 1 - \Theta(t\delta).$$

$\square$

By exactly the same as in Sec. C.3, we can lower bound the scaled gradient term by

$$\nabla\mathcal{L}(x_t)^\top \hat{V}_{t-1}^{-1/2}\nabla\mathcal{L}(x_t) \geq \min_i(\hat{V}_{t-1}^{-1/2})_i \sum_{i=1}^{d}[\nabla\mathcal{L}(x_t)]_i^2$$

$$\geq (\sqrt{1+\frac{\log^{1.5}(CKtd/\delta)}{\sqrt{b}}}\eta_t KG + \epsilon)^{-1}\|\nabla\mathcal{L}(x_t)\|^2, \;\; w.p.\; 1-\Theta(d\delta).$$

On the martingale of zero-centered noises, we can simply incorporate the learning rate $\eta_t$ into the martingale. Define the random process of sketching noise as

$$\{Y_t = \sum_{\tau=1}^{t}\frac{\eta_\tau}{C}\sum_{k=1}^{K}\nabla\mathcal{L}(x_\tau)^\top \hat{V}_{\tau-1}^{-1/2}(R_\tau^\top R_\tau g_{\tau,k}^c - g_{\tau,k}^c)\}_{t=1}^{T}$$

as a martingale. The difference of $|Y_t - Y_{t-1}|$ is bounded with high probability

$$|Y_t - Y_{t-1}| = |\frac{\eta_t}{C}\sum_{c=1}^{C}\sum_{k=1}^{K}\nabla\mathcal{L}(x_t)^\top \hat{V}_{t-1}^{-1/2}(R_t^\top R_t g_{t,k}^c - g_{t,k}^c)|$$

$$\leq \frac{\log^{1.5}(d/\delta)}{\sqrt{b}}\eta_t KG\|\hat{V}_t^{-1/2}\nabla\mathcal{L}(x_t)\|_2, \;\; w.p.\; 1-\Theta(CK\delta).$$

Then by Azuma's inequality,

$$\mathbb{P}(|Y_T| \geq \nu\sqrt{\sum_{t=1}^{T}\left(\frac{\log^{1.5}(d/\delta)}{\sqrt{b}}\eta_t KG\|\hat{V}_t^{-1/2}\nabla\mathcal{L}(x_t)\|_2\right)^2}) = O(\exp(-\Omega(\nu^2))) + T\delta \quad (5)$$

A similar bound can be achieved for the sub-Gaussian noise in stochastic gradient. Let

$$Z_t = \sum_{\tau=1}^{t}\frac{\eta_\tau}{C}\sum_{k=1}^{K}\nabla\mathcal{L}(x_\tau)^\top \hat{V}_{\tau-1}^{-1/2}(g_{\tau,k}^c - \nabla\mathcal{L}^c(x_{t,k}^c)).$$

Then

$$\mathbb{P}(|Z_T| \geq \nu\sqrt{\sum_{t=1}^{T}(\frac{\eta_t\sigma}{\epsilon})^2\log(\frac{2T}{\delta_g})}) = O(\exp(-\Omega(\nu^2))) + \delta_g$$

Combining the two bounds by union bound completes the proof.

# D   Convergence Without Bounded Gradient Norm Assumption (Proof of Theorem 3.4)

We first prove the local client gradient $\mathcal{L}^c$ is bounded. The client performs stochastic gradient descent $x_{t,k}^c = x_t - \eta\sum_{\tau=1}^{k}g_{t,\tau}^c$. Let $\eta = \frac{\eta_0}{\sqrt{K}}$

**Lemma D.1.** *Under Assumption 3, Let $\eta \leq \frac{1}{2L\sqrt{K}}$. The local gradients as of $k \leq K$ can be bounded with probability $1 - K\delta_c - K\exp(-\Delta^2/\sigma^2)$ by*

$$\|\nabla\mathcal{L}^c(x_{t,k}^c)\| \leq \sqrt{2\Delta^2\ln\frac{2}{\delta_c}} + \sqrt{2\Delta^2\ln\frac{2}{\delta_c} + 4L\mathcal{L}^c(x_t) + \Delta^2}.$$

*Proof.*

$$\frac{1}{2L}\|\nabla\mathcal{L}^c(x_{t,k}^c)\|^2 \leq \mathcal{L}^c(x_{t,k}^c) \leq \mathcal{L}^c(x_t) + \sum_{k=1}^{K}\langle\nabla\mathcal{L}^c(x_t), x_{t,k}^c - x_{t,k-1}^c\rangle + \frac{L}{2}\|x_{t,k}^c - x_{t,k-1}^c\|^2$$

$$= \mathcal{L}^c(x_t) - \eta\sum_{k=1}^{K}\langle\nabla\mathcal{L}^c(x_{t,k}^c), \nabla\mathcal{L}^c(x_{t,k}^c) + \epsilon_{t,k}^c\rangle + \frac{L}{2}\|x_{t,k}^c - x_{t,k-1}^c\|^2$$

$$= \mathcal{L}^c(x_t) + \eta\sum_{k=1}^{K}-\|\nabla\mathcal{L}^c(x_{t,k}^c)\|^2 - \eta\sum_{\tau=1}^{k-1}\langle\nabla\mathcal{L}^c(x_{t,\tau}^c), \epsilon_{t,\tau}^c\rangle + \eta^2\sum_{\tau=1}^{k}L(\|\nabla\mathcal{L}^c(x_{t,\tau}^c)\|^2 + \|\epsilon_{t,\tau}^c\|^2)$$

Take induction basis $\tau \leq k-1$. We have bounded gradient $\|\nabla\mathcal{L}^c(x_{t,\tau}^c)\|^2 \leq G$ with probability $1 - \tau\delta_c - \tau\exp(-\Delta^2/\sigma^2)$. The RHS can be bounded with probability $1 - k\delta_c - k\exp(-\Delta^2/\sigma^2)$ by

$$\mathcal{L}^c(x_t) - \eta\sum_{\tau=1}^{k-1}\langle\nabla\mathcal{L}^c(x_{t,\tau}^c), \epsilon_{t,\tau}^c\rangle + \eta^2\sum_{\tau=1}^{k-1}L(\|\nabla\mathcal{L}^c(x_{t,\tau}^c)\|^2 + \|\epsilon_{t,\tau}^c\|^2)$$

$$\leq \mathcal{L}^c(x_t) + \frac{\eta_0}{\sqrt{K}}\sqrt{2KG\Delta^2\ln\frac{2}{\delta_c}} + \frac{\eta_0^2 L}{K}K(G+\Delta^2)$$

$$\leq \mathcal{L}^c(x_t) + \eta_0\sqrt{2G\Delta^2\ln\frac{2}{\delta_c}} + \eta_0^2 L(G+\Delta^2)$$

$$\leq \mathcal{L}^c(x_t) + \frac{\eta_0}{2}G + \eta_0\Delta^2\ln\frac{2}{\delta_c} + \eta_0^2 L(G+\Delta^2) \leq \frac{G}{2L}$$

Let $\eta_0 \leq \frac{1}{2L}$, and $G = (\sqrt{2\Delta^2\ln\frac{2}{\delta_c}} + \sqrt{2\Delta^2\ln\frac{2}{\delta_c} + 4L\mathcal{L}^c(x_t) + \Delta^2})^2$, we have

$$\text{RHS} = \mathcal{L}^c(x_t) + \frac{\eta_0}{2}G + \eta_0\Delta^2\ln\frac{2}{\delta_c} + \eta_0^2 L(G + \Delta^2\ln\frac{2}{\delta_c})$$

$$\leq \mathcal{L}^c(x_t) + \frac{1}{4L}G + \frac{1}{2L}\Delta^2\ln\frac{2}{\delta_c} + \frac{1}{4L}(G + \Delta^2\ln\frac{2}{\delta_c}) = \frac{G}{2L}$$

$\square$

**Lemma 3.2.** *Under Assumption 3, Let $\eta \leq \frac{1}{2L\sqrt{K}}$. The local gradients as of $k \leq K$ can be bounded with probability $1 - CK\delta_c - CK\exp(-\Delta^2/\sigma^2)$ by*

$$\frac{1}{C}\sum_{c=1}^{C}\|\nabla\mathcal{L}^c(x_{t,\tau}^c)\| \leq 2\sqrt{L}\sqrt{\mathcal{L}(x_t)} + 2\sqrt{2\Delta^2\ln\frac{2}{\delta_c}} + \Delta.$$

*Proof.* Applying the fact that $\mathcal{L}(x^t) = \frac{1}{C}\sum_{c=1}^{C}\mathcal{L}^c(x_t)$ to Lemma D.1, the averaged local gradient can be bounded by the global loss,

$$\frac{1}{C}\sum_{c=1}^{C}\|\nabla\mathcal{L}^c(x_{t,\tau}^c)\| \leq 2\sqrt{L}\sqrt{\mathcal{L}(x_t)} + 2\sqrt{2\Delta^2\ln\frac{2}{\delta_c}} + \Delta.$$

The averaged local gradient norm will be a key component in the following analysis that focuses on the global gradients. $\square$

Consider the server optimizer

$$\mathcal{L}(z_{t+1}) = \mathcal{L}(z_t) + \nabla\mathcal{L}(z_t)^\top(z_{t+1} - z_t) + \frac{1}{2}(z_{t+1} - z_t)^\top\hat{H}_{\mathcal{L}}(z_{t+1} - z_t)$$

$$= \mathcal{L}(z_t) + \nabla\mathcal{L}(x_t)^\top(z_{t+1} - z_t) + (\nabla\mathcal{L}(z_t) - \nabla\mathcal{L}(x_t))^\top(z_{t+1} - z_t) + \frac{1}{2}(z_{t+1} - z_t)^\top\hat{H}_{\mathcal{L}}(z_{t+1} - z_t).$$

$$\nabla\mathcal{L}(x_t)^\top(z_{t+1} - z_t)$$

$$= \nabla\mathcal{L}(x_t)^\top\left(\frac{\beta_1}{1-\beta_1}\left(\kappa\hat{V}_{t-1}^{-1/2} - \kappa\hat{V}_t^{-1/2}\right)m_{t-1} - \frac{\kappa\eta}{C}\hat{V}_t^{-1/2}\sum_{c=1}^{C}\sum_{k=1}^{K}R_t^\top R_t g_{t,k}^c\right)$$

$$= \frac{\beta_1}{1-\beta_1}\nabla\mathcal{L}(x_t)^\top\left(\kappa\hat{V}_{t-1}^{-1/2} - \kappa\hat{V}_t^{-1/2}\right)m_{t-1}$$

$$- \frac{\kappa\eta}{C}\nabla\mathcal{L}(x_t)^\top\hat{V}_t^{-1/2}\sum_{c=1}^{C}\sum_{k=1}^{K}\nabla\mathcal{L}^c(x_t) - \nabla\mathcal{L}(x_t) + \nabla\mathcal{L}^c(x_{t,k}^c) - \nabla\mathcal{L}^c(x_{t,k}^c) + g_{t,k}^c - g_{t,k}^c + R_t^\top R_t g_{t,k}^c$$

$$\frac{1}{C}\sum_{c=1}^{C}\|\nabla\mathcal{L}^c(x_{t,\tau}^c)\| \leq \frac{1}{C}\sum_{c=1}^{C}\sqrt{2\Delta^2\ln\frac{2}{\delta_c}} + \sqrt{2\Delta^2\ln\frac{2}{\delta_c} + 4L\mathcal{L}^c(x_t) + \Delta^2}$$

$$\leq \frac{1}{C}\sum_{c=1}^{C}2\sqrt{L\mathcal{L}^c(x_t)} + 2\sqrt{2\Delta^2\ln\frac{2}{\delta_c}} + \Delta$$

$$\leq \frac{2\sqrt{L}}{C}\sqrt{C\sum_{c=1}^{C}\mathcal{L}^c(x_t)} + 2\sqrt{2\Delta^2\ln\frac{2}{\delta_c}} + \Delta$$

$$= 2\sqrt{L}\sqrt{\mathcal{L}(x_t)} + 2\sqrt{2\Delta^2\ln\frac{2}{\delta_c}} + \Delta$$

where the third inequality follows by Cauchy-Schwarz. On the server side, we consider the induction basis $\frac{1}{2L}\|\nabla\mathcal{L}(x_t)\|^2 \leq \mathcal{L}(x_t) \leq \frac{G}{2L}$, $w.p.$ $1 - t\exp(-\Omega(\nu^2)) - tCK\delta_c - tCK\exp(-\Delta^2/\sigma^2)$ holds for $t \leq T - 1$. The following inequality holds with probability $1 - K\delta_c - CK\exp(-\Delta^2/\sigma^2)$,

$$\frac{\kappa\eta}{C}\nabla\mathcal{L}(x_t)^\top\hat{V}_t^{-1/2}\sum_{c=1}^{C}\sum_{k=1}^{K}-\nabla\mathcal{L}^c(x_t) + \nabla\mathcal{L}^c(x_{t,k}^c)$$

$$\leq \frac{\kappa\eta}{C}\|\nabla\mathcal{L}(x_t)\|\|\hat{V}_t^{-1/2}\|\sum_{c=1}^{C}\sum_{k=1}^{K}\eta L\|\sum_{\tau=1}^{k-1}g_{t,\tau}^c\|$$

$$\leq \frac{\kappa\eta^2 L}{C}\|\nabla\mathcal{L}(x_t)\|\|\hat{V}_t^{-1/2}\|\sum_{c=1}^{C}\sum_{k=1}^{K}\|\sum_{\tau=1}^{k-1}g_{t,\tau}^c\|$$

$$\leq \frac{\kappa\eta^2 L}{\epsilon C}\|\nabla\mathcal{L}(x_t)\|\sum_{c=1}^{C}\sum_{k=1}^{K}\sum_{\tau=1}^{k-1}\|\nabla\mathcal{L}^c(x_{t,\tau}^c)\| + \Delta$$

$$\leq \frac{\kappa\eta^2 K^2 L}{\epsilon}\|\nabla\mathcal{L}(x_t)\|(2\sqrt{L}\sqrt{\mathcal{L}(x_t)} + 2\sqrt{2\Delta^2\ln\frac{2}{\delta_c}} + \Delta) + \frac{\kappa\eta^2 K^2 L}{\epsilon}\|\nabla\mathcal{L}(x_t)\|\Delta$$

$$\leq \frac{\sqrt{2}\kappa\eta^2 K^2 L^2}{\epsilon}\mathcal{L}(x_t) + \frac{2\kappa\eta^2 K^2 L}{\epsilon}\|\nabla\mathcal{L}(x_t)\|\Delta(1 + \sqrt{2\ln\frac{2}{\delta_c}})$$

And for the difference term, applying Lemma B.2 yields

$$\frac{\eta}{C}\nabla\mathcal{L}(x_t)^\top(\kappa\hat{V}_t^{-1/2} - \kappa\hat{V}_{t-1}^{-1/2})\sum_{c=1}^{C}\sum_{k=1}^{K}R_t^\top R_t g_{t,k}^c$$

$$\leq \frac{\eta\kappa}{C}(1 + \frac{\log^{1.5}(CKTd/\delta)}{\sqrt{b}})\|\nabla\mathcal{L}(x_t)\|\|\hat{V}_t^{-1/2} - \hat{V}_{t-1}^{-1/2}\|_2\sum_{c=1}^{C}\sum_{k=1}^{K}\|g_{t,k}^c\|$$

Denote $[\cdot]_i$ as the $i$-th element of a vector. The $l2$-norm

$$\|\hat{V}_t^{-1/2} - \hat{V}_{t-1}^{-1/2}\|_2 = \max_i \frac{1}{\sqrt{\hat{v}_{t-1,i}} + \epsilon} - \frac{1}{\sqrt{\hat{v}_{t,i}} + \epsilon} = \max_i \frac{\sqrt{\hat{v}_{t,i}} - \sqrt{\hat{v}_{t-1,i}}}{(\sqrt{\hat{v}_{t-1,i}} + \epsilon)(\sqrt{\hat{v}_{t,i}} + \epsilon)}$$

$$= \max_i \frac{\hat{v}_{t,i} - \hat{v}_{t-1,i}}{(\sqrt{\hat{v}_{t-1,i}} + \epsilon)(\sqrt{\hat{v}_{t,i}} + \epsilon)(\sqrt{\hat{v}_{t,i}} + \sqrt{\hat{v}_{t-1,i}})}$$

By definition, $\hat{v}_t = \max(\hat{v}_{t-1}, v_t)$. If $\hat{v}_{t,i} = \hat{v}_{t-1,i}$, the RHS is 0. Otherwise, $\hat{v}_{t,i} = v_{t,i}$.

$$\|\hat{V}_t^{-1/2} - \hat{V}_{t-1}^{-1/2}\|_2 \leq \max_i \frac{v_{t,i} - v_{t-1,i}}{(\sqrt{\hat{v}_{t-1,i}} + \epsilon)(\sqrt{\hat{v}_{t,i}} + \epsilon)(\sqrt{\hat{v}_{t,i}} + \sqrt{\hat{v}_{t-1,i}})}$$

$$\leq \max_i \frac{(1-\beta_2)(\bar{v}_{t,i} - v_{t-1,i})}{\epsilon^2 \sqrt{(1-\beta_2)\bar{v}_{t,i}}}$$

$$\leq \max_i \frac{\sqrt{1-\beta_2}}{\epsilon^2} \sqrt{\bar{v}_{t,i}}$$

$$= \frac{\sqrt{1-\beta_2}}{\epsilon^2} \max_i \sqrt{\frac{\eta^2}{C} \sum_{c=1}^{C} \sum_{k=1}^{K} [(\sum R_t^\top R_t g_{t,k}^c)^2]_i}$$

$$\leq \frac{\eta\sqrt{2(1-\beta_2)}}{\epsilon^2} \sqrt{1 + \frac{\log^{1.5}(CKtd^2/\delta)}{\sqrt{b}}} (\sqrt{2G} + 2\Delta(1 + \sqrt{2\ln \frac{2CK}{\delta_c}})).$$

The first inequality is from $\hat{v}_{t-1,i} \geq v_{t-1,i}$. The second inequality comes from $\hat{v}_{t,i} \geq v_{t,i} \geq (1-\beta_2)\bar{v}_{t,i}$. The last inequality follows from applying Lemma B.2 to each dimension of $g_{t,k}^c$. Plugging into the bound for the difference term

$$\frac{\eta}{C} \nabla\mathcal{L}(x_t)^\top (\kappa\hat{V}_t^{-1/2} - \kappa\hat{V}_{t-1}^{-1/2}) \sum_{c=1}^{C} \sum_{k=1}^{K} R_t^\top R_t g_{t,k}^c$$

$$\leq \frac{\eta^2 \kappa \sqrt{2(1-\beta_2)}}{\epsilon^2}(1 + \frac{\log^{1.5}(CKtd^2/\delta)}{\sqrt{b}})^{3/2}\sqrt{G}(\sqrt{2G} + 2\Delta(1 + \sqrt{2\ln \frac{2CK}{\delta_c}}))^2$$

Consider the sketching noise term. Since the noise is zero-centered, we view the random process of

$$\{Y_t = \sum_{\tau=1}^{t} \frac{1}{C} \sum_{c=1}^{C} \sum_{k=1}^{K} \nabla\mathcal{L}(x_\tau)^\top \hat{V}_{\tau-1}^{-1/2}(R_\tau^\top R_\tau g_{\tau,k}^c - g_{\tau,k}^c)\}_{t=1}^{T-1}$$

as a martingale. The difference of $|Y_{t+1} - Y_t|$ is bounded with high probability

$$|Y_{t+1} - Y_t| \leq \frac{1}{C} \sum_{c=1}^{C} \sum_{k=1}^{K} |\nabla\mathcal{L}(x_t)^\top \hat{V}_{t-1}^{-1/2}(R_t^\top R_t g_{t,k}^c - g_{t,k}^c)| \leq \sum_{k=1}^{K} \frac{\log^{1.5}(CKd/\delta)}{\sqrt{b}}\|g_{t,k}^c\|\|\hat{V}_{t-1}^{-1/2}\nabla\mathcal{L}(x_t)\|_2$$

$$\leq \frac{\log^{1.5}(CKd/\delta)}{\sqrt{b}} \frac{K}{\epsilon}\left(2\sqrt{2}L\mathcal{L}(x_t) + 2\|\nabla\mathcal{L}(x_t)\|\Delta(1 + \sqrt{2\ln \frac{2CK}{\delta_c}})\right)$$

Then by Azuma's inequality, with probability at least $1 - T\exp(-\Omega(\nu^2)) - T\delta_c - TCK\exp(-\Delta^2/\sigma^2)$

$$|Y_T| \leq \nu \frac{\log^{1.5}(CKTd/\delta)}{\sqrt{b}} \frac{K}{\epsilon}\left(\sum_{t=1}^{T}(2\sqrt{2}L\mathcal{L}(x_t) + 2\|\nabla\mathcal{L}(x_t)\|\Delta(1 + \sqrt{2\ln \frac{2CK}{\delta_c}}))^2\right)^{1/2}$$

$$\leq \nu \frac{\log^{1.5}(CKTd/\delta)}{\sqrt{b}} \frac{K}{\epsilon}\sqrt{T}(\sqrt{2G} + 2\sqrt{G}\Delta(1 + \sqrt{2\ln \frac{2CK}{\delta_c}}))$$

where the second inequality follows from the induction basis.

We also consider the product term $|(\hat{V}_t^{-1/2}m_t)^\top v_i|$.

**Lemma D.2.** *With probability* $1 - \Theta(t\delta)$*, for eigenvector* $v_i$ *of the Hessian matrix,* $|(\hat{V}_t^{-1/2}m_t)^\top v_i| \leq (1 + \frac{\log^{1.5}(CKd/\delta)}{\sqrt{b}})\eta KG/\epsilon$.

*Proof.* We can prove by induction. For $t = 0$, since $m_0 = 0$, the inequality holds. By the induction basis, $\|g_{t,k}^c\|$ has a uniform upper bound. Suppose we have for $h \in \mathbb{R}^d$, s.t. $\|h\| \leq H$, with probability $1 - \Theta((t-1)\delta)$,

$$|m_{t-1}^\top h| \leq (1 + \frac{\log^{1.5}(CKd/\delta)}{\sqrt{b}})\eta KH(\sqrt{2G} + 2\Delta(1 + \sqrt{2\ln \frac{2CK}{\delta_c}}))$$

Then by the update rule,

$$
|m_t^\top h| = |(\beta_1 \cdot m_{t-1} + (1-\beta_1) \cdot \frac{\eta}{C} \sum_{c=1}^{C} \sum_{k=1}^{K} R_t^\top R_t g_{t,k}^c)^\top h|
$$

$$
\leq \beta_1 |m_{t-1}^\top h| + \frac{(1-\beta_1)\eta}{C} \sum_{c=1}^{C} \sum_{k=1}^{K} |\langle R_t^\top R_t g_{t,k}^c, h \rangle|
$$

$$
\leq \beta_1 |m_{t-1}^\top h| + (1-\beta_1)(1 + \frac{\log^{1.5}(CKd/\delta)}{\sqrt{b}})\eta \frac{1}{C} \sum_{c=1}^{C} \sum_{k=1}^{K} \|g_{t,k}^c\|_2 \|h\|_2
$$

$$
\leq (1 + \frac{\log^{1.5}(CKd/\delta)}{\sqrt{b}})\eta KH(\sqrt{2G} + 2\Delta(1 + \sqrt{2\ln \frac{2CK}{\delta_c}})), \;\; w.p.\; 1 - \Theta(t\delta).
$$

Let $h = \hat{V}_t^{-1/2} v_i$. Then $\|h\|_2 \leq 1/\epsilon$. We have

$$
|(\hat{V}_t^{-1/2} m_t)^\top v_i| \leq (1 + \frac{\log^{1.5}(CKd/\delta)}{\sqrt{b}})\eta K(\sqrt{2G} + 2\Delta(1 + \sqrt{2\ln \frac{2CK}{\delta_c}}))/\epsilon
$$

$\square$

Then we consider the quadratic term, with probability $1 - t\delta - tCK \exp(-\Delta^2/\sigma^2)$

$$
(\nabla \mathcal{L}(z_t) - \nabla \mathcal{L}(x_t))^\top (z_{t+1} - z_t)
$$

$$
\leq \kappa^2 \frac{\beta_1}{(1-\beta_1)^2} (\hat{V}_{t-1}^{-1/2} m_{t-1})^\top \hat{H}_\mathcal{L}(\hat{V}_t^{-1/2} m_t) - \kappa^2 \frac{\beta_1^2}{(1-\beta_1)^2}(\hat{V}_{t-1}^{-1/2} m_{t-1})^\top \hat{H}_\mathcal{L}(\hat{V}_{t-1}^{-1/2} m_{t-1}),
$$

$$
= \kappa^2 \frac{\beta_1}{(1-\beta_1)^2} \sum_{i=1}^{d} \lambda_i (\hat{V}_{t-1}^{-1/2} m_{t-1})^\top (v_i v_i^\top)\hat{V}_t^{-1/2} m_t - \kappa^2 \frac{\beta_1^2}{(1-\beta_1)^2} \sum_{i=1}^{d} \lambda_i (\hat{V}_{t-1}^{-1/2} m_{t-1})^\top (v_i v_i^\top)\hat{V}_{t-1}^{-1/2} m_{t-1}
$$

$$
\leq \kappa^2 \frac{\beta_1}{(1-\beta_1)^2} \sum_{i=1}^{d} |\lambda_i| |(\hat{V}_{t-1}^{-1/2} m_{t-1})^\top v_i| |(\hat{V}_t^{-1/2} m_t)^\top v_i| + \kappa^2 \frac{\beta_1^2}{(1-\beta_1)^2} \sum_{i=1}^{d} |\lambda_i| |(\hat{V}_{t-1}^{-1/2} m_{t-1})^\top v_i|^2
$$

$$
\leq \kappa^2 \frac{2}{(1-\beta_1)^2} \mathcal{I}L(1 + \frac{\log^{1.5}(CKd^2/\delta)}{\sqrt{b}})^2 \eta^2 K^2 (\sqrt{2G} + 2\Delta(1 + \sqrt{2\ln \frac{2CK}{\delta_c}}))^2/\epsilon^2
$$

$$
\leq \kappa^2 \frac{8}{(1-\beta_1)^2} \mathcal{I}L(1 + \frac{\log^{1.5}(CKd^2/\delta)}{\sqrt{b}})^2 \eta^2 K^2 (G + 2\Delta^2(1 + \sqrt{2\ln \frac{2CK}{\delta_c}})^2)/\epsilon^2
$$

where the last but one inequality is by $\beta_1 \leq 1$ and Lemma. D.2.

Putting all these things together, with probability $1 - T\exp(-\Omega(\nu^2)) - TC\delta_c - TCK\exp(-\Delta^2/\sigma^2) - T\delta$

$$
\mathcal{L}(x_T) = \mathcal{L}(z_T) + \frac{\beta_1}{1-\beta_1}\langle\nabla\mathcal{L}(z_T), \kappa\hat{V}_t^{-1/2}m_t\rangle + \frac{\kappa^2}{2}\frac{\beta_1^2}{(1-\beta_1)^2}(\hat{V}_t^{-1/2}m_t)^\top H_\mathcal{L}(\hat{V}_t^{-1/2}m_t)
$$

$$
= \mathcal{L}(z_1) + \sum_{t=1}^{T-1}\nabla\mathcal{L}(x_t)^\top(z_{t+1}-z_t) + (\nabla\mathcal{L}(z_t)-\nabla\mathcal{L}(x_t))^\top(z_{t+1}-z_t) + \frac{1}{2}(z_{t+1}-z_t)^\top\hat{H}_\mathcal{L}(z_{t+1}-z_t)
$$

$$
+ \frac{\beta_1}{1-\beta_1}\langle\nabla\mathcal{L}(z_T), \kappa\hat{V}_t^{-1/2}m_t\rangle + \frac{\kappa^2}{2}\frac{\beta_1^2}{(1-\beta_1)^2}(\hat{V}_t^{-1/2}m_t)^\top H_\mathcal{L}(\hat{V}_t^{-1/2}m_t)
$$

$$
\leq 2\mathcal{L}(z_1) + \frac{8\eta\beta_1}{(1-\beta_1)\epsilon} + \sum_{t=1}^{T-1}\frac{2\sqrt{2}\kappa\eta^2 K^2 LG}{\epsilon} + \frac{4\kappa\eta^2 K^2 L\sqrt{G}}{\epsilon}\Delta(1+\sqrt{2\ln\frac{2CK}{\delta_c}})
$$

$$
+ \sum_{t=1}^{T-1}\frac{2\eta^2\kappa\sqrt{2(1-\beta_2)}}{\epsilon^2}(1+\frac{\log^{1.5}(CKtd^2/\delta)}{\sqrt{b}})^{3/2}\sqrt{G}(\sqrt{2G}+2\Delta(1+\sqrt{2\ln\frac{2CK}{\delta_c}}))^2
$$

$$
+ 2\kappa\eta\nu\frac{\log^{1.5}(CKTd/\delta)}{\sqrt{b}}\frac{K}{\epsilon}\sqrt{T}(\sqrt{2}G+2\sqrt{G}\Delta(1+\sqrt{2\ln\frac{2CK}{\delta_c}})) + 2\kappa\eta\nu\sqrt{T}\frac{K\sqrt{G}}{\epsilon}\Delta
$$

$$
+ \sum_{t=1}^{T-1}\kappa^2\frac{16}{(1-\beta_1)^2}\mathcal{I}L(1+\frac{\log^{1.5}(CKd^2/\delta)}{\sqrt{b}})^2\eta^2 K^2(G+2\Delta^2(1+\sqrt{2\ln\frac{2CK}{\delta_c}})^2)/\epsilon^2
$$

$$
+ \sum_{t=1}^{T-1}\left(\frac{1+\beta_1}{1-\beta_1}\right)^2(1+\frac{\log^{1.5}(CKd^2/\delta)}{\sqrt{b}})^2 2\kappa^2\eta^2 K^2\mathcal{I}L(G+2\Delta^2(1+\sqrt{2\ln\frac{2CK}{\delta_c}})^2)/\epsilon^2
$$

$$
\leq 2\mathcal{L}(z_1) + \frac{8\eta\beta_1}{(1-\beta_1)\epsilon} + \frac{\kappa\eta_0^2 K^2 L\sqrt{G}}{\epsilon}4\sqrt{2G} + \frac{\eta_0^2\kappa\sqrt{2(1-\beta_2)}}{\epsilon^2}(1+\frac{\log^{1.5}(CKtd^2/\delta)}{\sqrt{b}})^{3/2}16G^{3/2}
$$

$$
+ \kappa\eta_0\nu\frac{\log^{1.5}(CKTd/\delta)}{\sqrt{b}}\frac{K}{\epsilon}4\sqrt{2}G + 2\kappa\eta_0\nu\frac{K\sqrt{G}}{\epsilon}\Delta
$$

$$
+ \kappa^2\frac{8+(1+\beta_1)^2}{(1-\beta_1)^2}\mathcal{I}L(1+\frac{\log^{1.5}(CKd^2/\delta)}{\sqrt{b}})^2 4\eta_0^2 K^2 G/\epsilon^2
$$

$$
\leq 2\mathcal{L}(z_1) + \frac{8\eta\beta_1}{(1-\beta_1)\epsilon} + \frac{\eta_0^2 K^2 L}{\epsilon}4\sqrt{2G} + \frac{\eta_0^2\sqrt{2(1-\beta_2)}}{\epsilon^2}(1+\frac{\log^{1.5}(CKtd^2/\delta)}{\sqrt{b}})^{3/2}16G
$$

$$
+ \eta_0\nu\frac{\log^{1.5}(CKTd/\delta)}{\sqrt{b}}\frac{K}{\epsilon}4\sqrt{2G} + 2\eta_0\nu\frac{K}{\epsilon}\Delta
$$

$$
+ \frac{8+(1+\beta_1)^2}{(1-\beta_1)^2}\mathcal{I}L(1+\frac{\log^{1.5}(CKTd^2/\delta)}{\sqrt{b}})^2 4\eta_0^2 K^2/\epsilon^2
$$

where the second inequality holds by $\sqrt{2G} \geq 2\Delta(1+\sqrt{2\ln\frac{2CK}{\delta_c}})$, the third inequality holds by $\kappa \leq \frac{1}{\sqrt{G}}$. Let

$$
\eta_0 \leq \frac{\epsilon}{2\sqrt{L}}\min\{\frac{1}{3}, \frac{1-\beta_1}{2\beta_1\sqrt{L}}\}(1+\frac{\log^{1.5}(CKTd^2/\delta)}{\sqrt{b}})^{-1}
$$

$$
G \geq \max\{2\Delta^2(1+\sqrt{2\ln\frac{2CK}{\delta_c}})^2, 512(\frac{\eta_0^2 K^2 L^2}{\epsilon} + \frac{\log^{1.5}(CKTd/\delta)}{\sqrt{b}}\frac{\eta_0\nu K}{\epsilon})^2 + 32L(\mathcal{L}(z_1)
$$

(6)

$$
+ \frac{4\eta\beta_1}{(1-\beta_1)\epsilon} + \frac{\eta_0\nu K\Delta}{\epsilon} + \frac{8+(1+\beta_1)^2}{(1-\beta_1)^2}(1+\frac{\log^{1.5}(CKTd^2/\delta)}{\sqrt{b}})^2\frac{2\eta_0^2 K^2\mathcal{I}L}{\epsilon^2})\} \quad (7)
$$

suffice to yield RHS $\leq \frac{G}{2L}$.

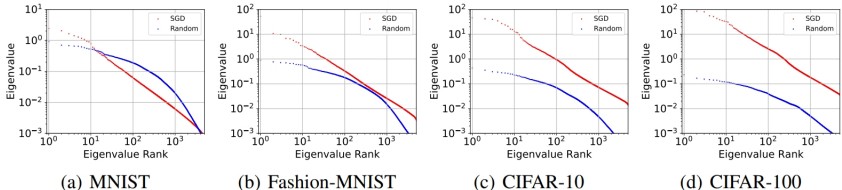

(a) MNIST     (b) Fashion-MNIST     (c) CIFAR-10     (d) CIFAR-100

Figure 4: The power-law structure of the Hessian spectrum on LeNet. Quoted from Fig.1 (Xie et al., 2022).

Furthermore, the dropped positive terms regarding the gradient norm is

$$\sum_{t=1}^{T} \kappa \eta K \nabla \mathcal{L}(x_t)^\top \hat{V}_t^{-1/2} \nabla \mathcal{L}(x_t)$$

$$\geq \kappa \eta_0 \sqrt{T} L \left( \sqrt{1 + \frac{\log^{1.5}(CKd^2T^2/\delta)}{\sqrt{b}}} \eta K(\sqrt{2G} + 2\Delta(1 + \sqrt{2\ln\frac{2}{\delta_c}})) + \epsilon \right)^{-1} \|\nabla \mathcal{L}(x_t)\|^2.$$

Rearranging the terms yields the convergence result.

Finally, we give the full forms of $\{\mathcal{M}_i\}_{i=1}^7$,

$$\mathcal{M}_1 := 2\sqrt{2}\nu \frac{\log^{1.5}(CKTd/\delta)}{\sqrt{b}} \frac{K}{\epsilon}$$

$$\mathcal{M}_2 := 4\sqrt{2}\nu \frac{\log^{1.5}(CKTd/\delta)}{\sqrt{b}} \frac{K}{\epsilon} 2\sqrt{G}\Delta(1 + \sqrt{2\ln\frac{2CK}{\delta_c}}) + 2\nu \frac{K}{\epsilon}\Delta$$

$$\mathcal{M}_3 := 2\mathcal{L}(z_1) + \frac{8\eta\beta_1}{(1-\beta_1)\epsilon}$$

$$\mathcal{M}_4 := \frac{4\sqrt{2(1-\beta_2)}}{\epsilon^2}(1 + \frac{\log^{1.5}(CKtd^2/\delta)}{\sqrt{b}})^{3/2}$$

$$\mathcal{M}_5 := \frac{2\sqrt{2}K^2 L}{\epsilon}$$

$$\mathcal{M}_6 := \frac{4K^2 L}{\epsilon}\Delta(1 + \sqrt{2\ln\frac{2CK}{\delta_c}}) + \frac{4\eta^2\kappa\sqrt{2(1-\beta_2)}}{\epsilon^2}(1 + \frac{\log^{1.5}(CKtd^2/\delta)}{\sqrt{b}})^{3/2}\ln\frac{2CK}{\delta_c}$$

$$\mathcal{M}_7 := 4\frac{8 + (1+\beta_1)^2}{(1-\beta_1)^2}\mathcal{I}(1 + \frac{\log^{1.5}(CKd^2/\delta)}{\sqrt{b}})^2 K^2/\epsilon^2$$

# E  Experimental Details and Additional Results

Aside from the experimental configurations described in the main paper, we provide additional details. We use Cross Entropy with label smoothing as the loss function. The parameter for label smoothing is 0.1. We use a cosine learning rate scheduler on the server optimizer, with the minimal learning rate is $1e-5$. Client batch size is 128, and weight decay is $1e-4$. We sweep a wide range of server learning rates over the set $[10^{-4}, 5 \times 10^{-4}, 10^{-3}, 5 \times 10^{-3}, 10^{-2}, 5 \times 10^{-2}, 0.1, 0.5, 1.0, 2.0]$. The algorithms share all the other hyper-parameters within each set of experiments. The only exception is CocktailSGD since it's unclear how to incorporate server-side momentum in their framework. To ensure a fair comparison, we tune both the server learning rate and the client side momentum for CocktailSGD. The selected hyperparameters are displayed in Table 2-7. We select the hyperparameters to balance between the training stability and model performance.

Our experiments were conducted on a computing cluster with AMD EPYC 7713 64-Core Processor and NVIDIA A100 Tensor Core GPU.

We provide the full comparison of the baseline methods on the benchmarks in Fig. 6 and Fig. 7. Although CocktailSGD and FedCAMS can achieve comparable performance under compression rate

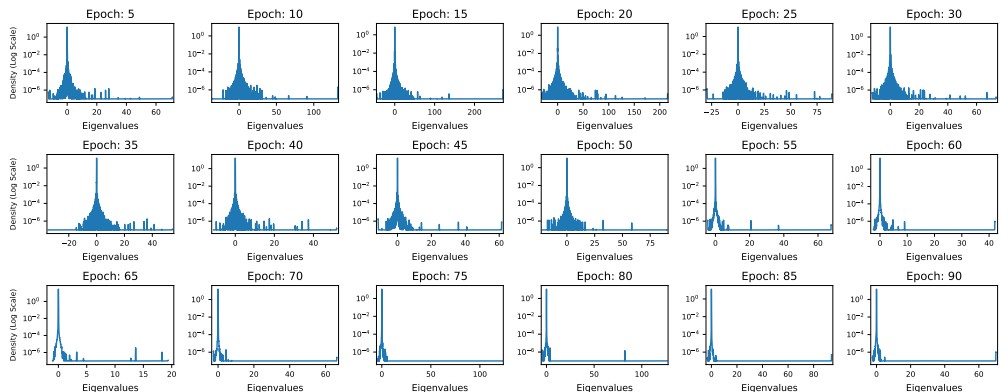

Figure 5: Eigenspectrum density every 5 epochs. The model is ViT-Small and trained on CIFAR10. The majority of eigenvalues concentrates near 0 and the density enjoys a super fast decay with the absolute values of eigenvalues, indicating a summable eigenspectra.

| Hyperparameter | SADL | CDAdam | FetchSGD | MARINA |
|---|---|---|---|---|
| Server learning rate | 0.01 | 0.01 | 1.0 | 0.1 |
| Server momentum | 0.9 | 0.9 | 0.9 | 0.9 |
| Server weight decay | 0.0001 | 0.0001 | 0.0001 | 0.0001 |
| Client learning rate | 0.1 | 0.1 | 0.1 | 0.1 |
| Client momentum | 0.0 | 0.0 | 0.0 | 0.0 |

Table 2: Hyperparameters of SADL, CDAdam, FetchSGD and MARINA in ResNet.

| Hyperparameter | CocktailSGD | CAMS | 1bit-Adam | PAQ |
|---|---|---|---|---|
| Server learning rate | 0.04 | 0.01 | 0.0001 | 0.1 |
| Server momentum | 0.0 | 0.9 | 0.9 | 0.9 |
| Server weight decay | 0.0 | 0.0001 | 0.0001 | 0.0001 |
| Client learning rate | 0.1 | 0.1 | 0.1 | 0.1 |
| Client momentum | 0.9 | 0.0 | 0.0 | 0.0 |

Table 3: Hyperparameters of CocktailSGD, CAMS, 1bit-Adam and PAQ in ResNet.

| Hyperparameter | SADL | CDAdam | FetchSGD | MARINA |
|---|---|---|---|---|
| Server learning rate | 0.001 | 0.0001 | 0.5 | 1.0 |
| Server momentum | 0.9 | 0.9 | 0.9 | 0.9 |
| Server weight decay | 0.0001 | 0.0001 | 0.0001 | 0.0001 |
| Client learning rate | 0.001 | 0.001 | 0.001 | 0.001 |
| Client momentum | 0.0 | 0.0 | 0.0 | 0.0 |

Table 4: Hyperparameters of SADL, CDAdam, FetchSGD and MARINA in SST.

| Hyperparameter | CocktailSGD | CAMS | 1bit-Adam | PAQ |
|---|---|---|---|---|
| Server learning rate | 0.001 | 0.001 | 0.00001 | 0.001 |
| Server momentum | 0.0 | 0.9 | 0.9 | 0.9 |
| Server weight decay | 0.0 | 0.0001 | 0.0001 | 0.0001 |
| Client learning rate | 0.001 | 0.001 | 0.001 | 0.001 |
| Client momentum | 0.9 | 0.0 | 0.0 | 0.0 |

Table 5: Hyperparameters of CocktailSGD, CAMS, 1bit-Adam and PAQ in SST.

| Hyperparameter | SADL | CDAdam | FetchSGD | MARINA |
|---|---|---|---|---|
| Server learning rate | 0.005 | 0.005 | 0.5 | 1.0 |
| Server momentum | 0.9 | 0.9 | 0.9 | 0.9 |
| Server weight decay | 0.005 | 0.005 | 0.005 | 0.005 |
| Client learning rate | 0.01 | 0.01 | 0.01 | 0.01 |
| Client momentum | 0.9 | 0.9 | 0.9 | 0.0 |

Table 6: Hyperparameters of SADL, CDAdam, FetchSGD and MARINA in ViT.

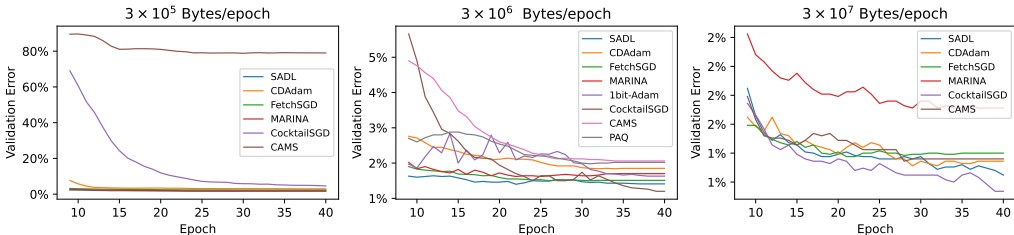

Figure 6: Validation Error on CIFAR-10. We finetune a ViT-base model (with 86M parameters) from the pretrained backbone checkpoint (Dosovitskiy et al., 2020). 1 bit-Adam has comparable communication cost with $3 \times 10^6$. SADL shows competitive performance under all communication budgets.

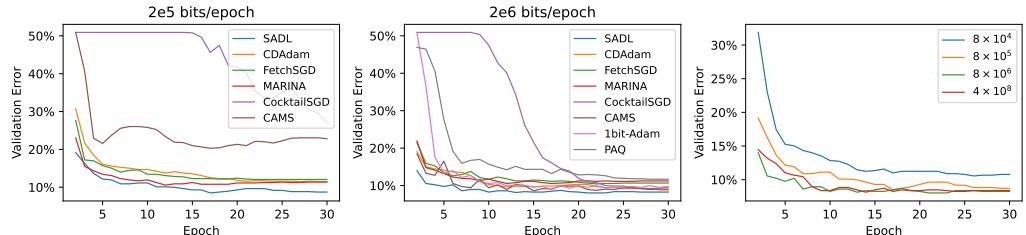

Figure 7: Validation Error on SST2 (GLUE) with BERT of 100M parameters. Left: compression rate $0.2\%$; Middle: $2\%$; Right: SADL with communication costs $\{8 \times 10^4, 8 \times 10^5, 8 \times 10^6, 4 \times 10^8\}$ Bytes/epoch. The legend $4 \times 10^8$ represents training in the ambient dimension without sketching. Higher compression rate improves the convergence rate and all compression rates achieve comparable test errors at the end of training.

$1\%$, their performance significantly degrades when the compression rate is lowered to $0.1\%$. On the other hand, our SADL framework has a consistent performance across different communication budgets. Notably, we observe that all algorithms, including SADL and the baselines, exhibit stable behavior during training. We report the statistical significance level of the ResNet experiment in the following table. Each experiment is repeated three times under different random seeds, with sources of randomness including client data partitioning, data shuffling, and sketching. Table 8 presents the validation error during the last communication round, with the values after $\pm$ indicating $3 \times$ standard deviation across the runs. 1bit-Adam and PAQ has a fixed compression rate, which amounts to $1.6 \times 10^6$ Bytes/epoch.

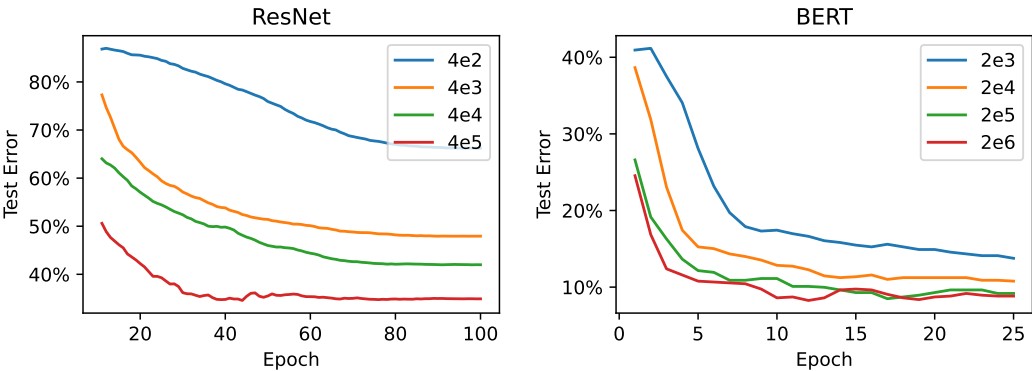

Figure 8: Comparing the performance of tiny sketch sizes on ResNet and BERT. The experiment settings are the same as in Fig. 1 and Fig. 3. The legends represent sketch sizes. In principle, an extremely tiny sketch size (with 400 in vision tasks and 2000 in language tasks) still converges at a similar rate but generates an unfavorable local minima that hardly generalizes.
.

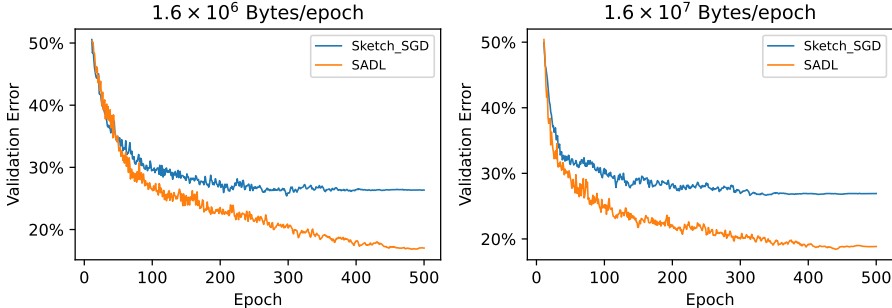

Figure 9: Validation Error on CIFAR-10. Comparing Sketch-SGD and SADL. The experimental settings are the same as in Fig. 1. SADL achieves faster convergence and better model performance.

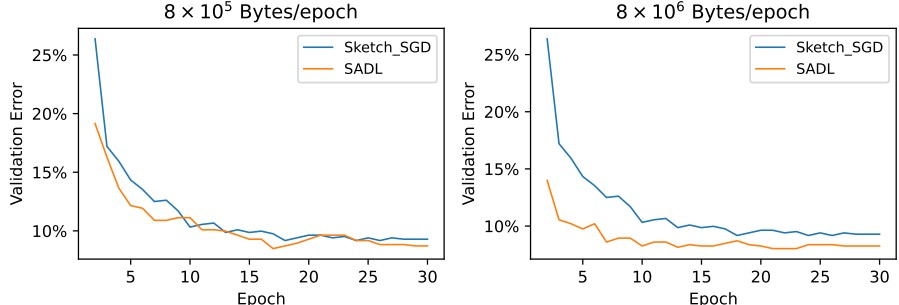

Figure 10: Validation Error on SST. Comparing Sketch-SGD and SADL. The experimental settings are the same as in Fig. 3. SADL achieves faster convergence and better model performance.

Next, we present the empirical comparison on using different server optimizers. We present the comparison on the three benchmarks in Fig. 9, 10 and 11. The sketch-SGD algorithm is the same as proposed in Song et al. (2023); Shrivastava et al. (2024). We show that using Adam as the server optimizer consistently outperforms in all communication budgets. We also provide the experimental results of running AMSGrad on all three experimental settings in the paper. Table 9, 10 and 11 present the validation errors during the last communication round, with the values after $\pm$ indicating $3\times$standard deviation across three independent runs. We can observe that Adam and AMSGrad has comparable performance on the all experimental settings, indicating that SADL is insensitive to the choice of optimizers.

Additionally, we include a more challenging set of experiments that considers data heterogeneity and a larger number of clients in Fig. 12. We train a ViT-Base model on CIFAR100 dataset. The FL system involves 30 clients (the number is maximized for error-feedback based methods under hardware constraints). The data is distributed in a heterogeneous way under Dirichlet distribution with Dirichlet prior $\alpha = 0.1$. We consider the compression rate 0.1% and 1%, as in the main paper. The validation accuracy along the training process is reported in Fig. 12. It can be observed that SADL achieves the best validation accuracy and converges fast.

To verify Assumption 4, we plot the full Hessian eigenspectrum throughout the training process in Fig. 5. We used stochastic lanczos algorithm implemented by the pyHessian library Yao et al. (2020) to approximate the distribution of the full eigenspectrum. Our main claim in Assumption 4 is that the

| Hyperparameter | CocktailSGD | CAMS | 1bit-Adam | PAQ |
|---|---|---|---|---|
| Server learning rate | 0.0001 | 0.001 | 0.0001 | 0.001 |
| Server momentum | 0.0 | 0.9 | 0.9 | 0.9 |
| Server weight decay | 0.0 | 0.005 | 0.005 | 0.005 |
| Client learning rate | 0.01 | 0.01 | 0.01 | 0.01 |
| Client momentum | 0.9 | 0.0 | 0.0 | 0.0 |

Table 7: Hyperparameters of CocktailSGD, CAMS, 1bit-Adam and PAQ in ViT.

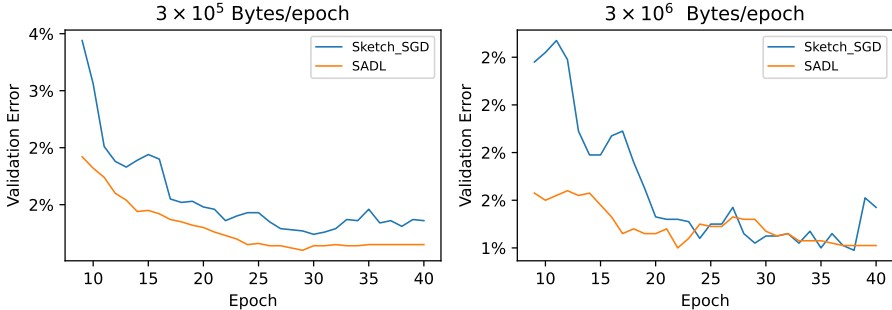

Figure 11: Validation Error on fine-tuning ViT with CIFAR-10 dataset. Comparing Sketch-SGD and SADL. The experimental settings are the same as in Fig. 2. SADL achieves faster convergence and comparable performance.

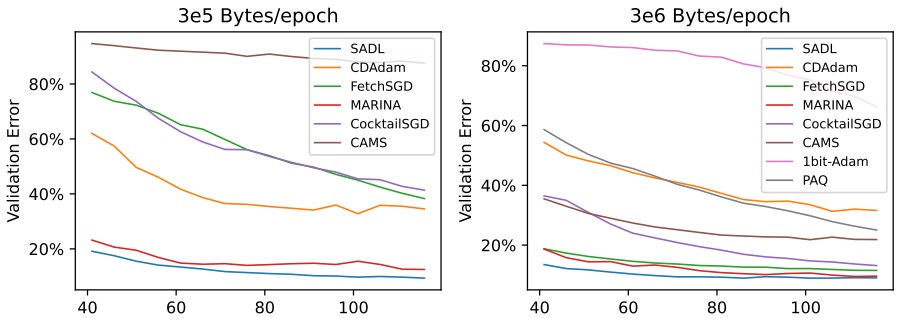

Figure 12: Validation Error of fine-tuning ViT-base (with 86M parameters) with CIFAR-100 dataset trained among 30 clients with heterogeneous data distribution. Left: The communication budget is $3 \times 10^5$ Bytes/epoch (0.1% compression rate). Right: The communication budget is $3 \times 10^6$ Bytes/epoch (1% compression rate). SADL shows competitive performance under all communication budgets.

Hessian eigenspectrum at an iterate is summable and the sum is independent of the ambient dimension, which can be satisfied by common distributions, like power-laws. We run testing experiments on ViT-small and train on CIFAR-10 dataset, with sketched Adam optimizer. In Fig. 5, we see the majority of eigenvalues concentrates near 0. The density enjoys a super fast decay with the absolute values of eigenvalues. The decay also holds throughout the training process. This empirical evidence shows the validity of our assumption.

In the main body of the paper, we have achieved 99.9% compression rate and 99.98% compression rate for ResNet and BERT respectively. We further include the results on smaller $b$ in Fig. 8. In principle, an extremely tiny sketch size (with 400 in vision tasks and 2000 in language tasks) still converges but generates an unfavorable local minima that hardly generalizes.

| Bytes/epoch | $1.6 \times 10^5$ | $1.6 \times 10^6$ |
|---|---|---|
| SADL | $(17.8 \pm 2.0)\%$ | $(16.9 \pm 0.8)\%$ |
| CDAdam | $(27.7 \pm 3.1)\%$ | $(25.4 \pm 1.9)\%$ |
| CAMS | $(41.2 \pm 4.9)\%$ | $(26.8 \pm 1.0)\%$ |
| CocktailSGD | $(47.4 \pm 2.3)\%$ | $(31.1 \pm 1.7)\%$ |
| MARINA | $(24.4 \pm 2.8)\%$ | $(24.8 \pm 4.1)\%$ |
| FetchSGD | $(36.2 \pm 2.4)\%$ | $(28.2 \pm 3.1)\%$ |
| 1bit-Adam | N/A | $(32.3 \pm 16.8)\%$ |
| PAQ | N/A | $(24.6 \pm 1.7)\%$ |

Table 8: Statistical Significance. The standard deviations are computed from 3 independent runs on the ResNet experiment.

| Bytes/epoch | $1.6 \times 10^5$ | $1.6 \times 10^6$ |
|---|---|---|
| Adam | $(17.8 \pm 2.0)\%$ | $(16.9 \pm 0.8)\%$ |
| AMSGrad | $(18.6 \pm 1.5)\%$ | $(17.3 \pm 2.1)\%$ |

Table 9: Comparing validation errors of AMSGrad and Adam as server optimizer in SADL on the ResNet experiment.

| Bytes/epoch | $3 \times 10^5$ | $3 \times 10^6$ |
|---|---|---|
| Adam | $(2.1 \pm 0.1)\%$ | $(1.6 \pm 0.1)\%$ |
| AMSGrad | $(1.8 \pm 0.3)\%$ | $(1.6 \pm 0.3)\%$ |

Table 10: Comparing validation errors of AMSGrad and Adam as server optimizer in SADL on the ViT experiment.

Additionally, we summarize the theoretical guarantees of the existing approaches in Table 12. From the table, we can see all the comparisons made in the main paper are fair.

## F   Discussions

**Limitations.** The major focus of this work is on distributed deep learning, which is a subset of the broader distributed learning setting. The specific scenario is of particular interest, since the models involved in modern deep learning are extremely large (typically in billions of parameters), leading to prohibitively high communication costs. Our theoretical analysis is based on the specific geometric structure in distributed deep learning. In extremely pessimistic cases or conventional optimization problems, the benefits of combining sketching and adaptive optimization require further analysis.

**Broader Impacts.** This work will facilitate distributed learning in achieving higher communication efficiency. The proposed SADL framework is especially beneficial for resource-constrained scenarios and contributes to more accessible distributed learning systems. However, the deployment of distributed training may also incur malicious attacks and data poisoning, which will require more attention in the future.

| Bytes/epoch | $8 \times 10^5$ | $8 \times 10^6$ |
|---|---|---|
| Adam | $(8.7 \pm 0.2)\%$ | $(8.1 \pm 0.3)\%$ |
| AMSGrad | $(8.8 \pm 0.2)\%$ | $(8.7 \pm 0.3)\%$ |

Table 11: Comparing validation errors of AMSGrad and Adam as server optimizer in SADL on the SST experiment.

| Algorithms | Communication Bits | learning rate | Convergence Rate |
|---|---|---|---|
| FetchSGD | $\tilde{O}(1)$ | $O(1/\sqrt{T})$ | $O(1/\sqrt{T})$ $^{(A)}$ |
| CocktailSGD | $O(1)$ | $O(1/(\sqrt{T} + T^{1/3}d^2 + d^3))$ | $O(1/\sqrt{T} + d^2/(T)^{2/3})$ |
| CD-Adam | $O(1)$ | $O(1/\sqrt{d})$ | $O(\sqrt{d}/\sqrt{T})$ |
| Onebit-Adam | $O(d)$ | $O(1/\sqrt{T})$ | $O(1/\sqrt{T})$ |
| MARINA | $O(1)$ | $(1 + \sqrt{\omega(d-b)/(bC)})^{-1}$ | $O(\sqrt{\frac{\omega}{n}(\frac{d}{b}-1)/T})$ $^{(B)}$ |
| FedCAMS | $O(1)$ | $O(1/\sqrt{T})$ | $O(d/\sqrt{T})$ |
| PAQ | $O(d)$ | $O(1/\sqrt{T})$ | $O(1/\sqrt{T})$ |
| Ours | $\tilde{O}(1)$ | $O(1/\sqrt{T})$ | $O(1/\sqrt{T})$ $^{(C)}$ |

Table 12: Comparison on Theoretical Guarantees. We only include the dependence on $d$ and $T$. (A) Needs a heavy-hitter assumption, otherwise deteriorated to $O(T^{1/3})$. (B) The rate is achieved either under deterministic case or use variance reduction methods. $\omega$ is typically $\Theta(d/b)$ when the compressor is RandK or $l_2$−quantization. (C) requires the assumption on the fast-decay Hessian eigenspectrum. Otherwise, the convergence rate can deteriorate to $O(d/\sqrt{T})$ under dimension-independent learning rate.

