# OpenReview forum: "Sketched Adaptive Distributed Deep Learning: A Sharp Convergence Analysis"
_NeurIPS.cc/2025/Conference — NeurIPS 2025 poster_

### Official Review · Reviewer_UgYH · 2025-06-21

**Clarity:** 3
**Significance:** 3
**Originality:** 3
**Rating:** 5
**Confidence:** 2

**Summary:**

This paper investigates sharper bounds for sketching for distributed optimization, under AMSGrad. A key contribution is to avoid reliance on d, the ambient dimension, and instead reduce to log d. This critically depends on the assumption of low intrinsic dimension, which is reasonable for many large DNNs.

**Questions:**

- the "third source of noise", is that really noise or bounding error? is it in practice or purely in the proof?

 - The idea of the limited intrinsic dimension is interesting, and I agree that it is likely true in practice. But can you discuss how baseline / competing sketching bounds compare to yours, if they too adopted that assumption? How much is this lack of reliance on $d$ a direct result of this assumption?

**Ethical Concerns:**

["NO or VERY MINOR ethics concerns only"]

**Final Justification:**

Looks good, I'll keep my score. Thanks for the discussion, it was very interesting to read.

**Limitations:**

yes

**Quality:**

3

**Strengths And Weaknesses:**

This paper seems to have substantial overlap with Song et al 2023; e.g. a lot of key theorems, and even the overall philosophy, seem borrowed from it. I guess this paper gives the novelty of investigating AMSGrad rather than gradient descent, which is significant, but my comment is that there should be a deeper discussion/review of this prior paper in the introduction, so that this novelty is clear. It also helps justify why the authors are so obsessed with AMSGrad, by showing that the GD method was already previously studied.

Overall, on a surface level, it seems a good contribution, although I will rely on others to comment on novelty.

More comments below:

It would be good ot have a more extensive discussion of the baseline algorithms


In terms of mathematical correctness, there were too many pages to check everything carefully. I did look through C1 and immediately found a few (likely minor) mistakes, e.g.

 - line 897: sign error in middle term
 - line 898: disappearing $\kappa V_t^{-1/2} m_{t-1}$ term, extra $R_t$ in front of $g_{t,k}^c$ (sketched variable and gradient should have same dimension)
 - line 900: missing nabla (definitely minor)

However, overall, the claims seem reasonable and I did not find anything too damaging in the overall logical flow of the proofs.


typos:
 - alg 1: parameters
 - Line 339: capitalization
 - 846: spacing
- Line 867: spacing

Although it is ok to give the results of Song et al 2023, it would be good to specify clearly the constants hidden in the $\Theta$ rate.

---

> ### Author Rebuttal · Authors · 2025-07-31
>
> We would like to extend our sincere appreciation to the reviewer for all the constructive and positive feedback on our contribution. We are especially grateful that the reviewer acknowledges the significance of our theoretical advancements. We would like to provide detailed responses regarding the weakness part and questions.
>
> - **Q1: Relationship with (Song et al. 2023).**
>
> Song et al. 2023 is an inspiring work that introduces the sketch-and-desketch framework to distributed learning. Our algorithm is a natural extension of the framework with **adaptive optimization**, with necessary adaptations to incorporate sketching in the precondtioners. Our theoretical analysis is grounded in the properties of standard sketching methods, as also outlined in Song et al 2023.
>
> However, the convergence bound derived in Song et al. is limited since _their communication cost scales **linearly** with the number of model parameters $d$_, i.e., $O(d)$, which is also there in prior results using other compression algorithms.
>
> The major advance in our work is to show an **exponential improvement** in the communication cost from $O(d)$ to $O(\log d)$, i.e., logarithmic in the number of parameters. The improvement is especially important for modern deep learning models with large number of parameters.
>
> Deriving the theoretical result in the context of _adaptive_ distributed learning is technically challenging, which requires more careful analysis on the effect of sketching in both the precondtioner term and noise terms. For the discussion on the preconditer term, please refer to our response to Q2 (below). For the analysis on noise terms, which is also not covered in Song et al 2023 or prior works, please refer to our response to Q3 (below).
>
> We will add a discussion in the introduction to properly give credit to Song et al., 2023 and highlight the connection and difference between our work and theirs.
>
>
> - **Q2: Significance of analyzing adaptive methods.**
>
> The use of adaptive optimizers poses unique challenges in leveraging the **anisotropic** structure (Definition 2.1) of deep learning optimization in the analysis. Adaptive optimizers naturally re-scales the update step to be **isotropic** over the ambient dimension via preconditioning. Our sharp analysis shows that the adaptive steps preserve the structure of the sketching noise so that the variance term can be bounded (Lemma 2.4).
>
> - **Q3: Does the intrinsic dimensionality suffice to derive the convergence bound?**
>
> Although intrinsic dimension is one of the key ingredients to derive our convergence bound, we would like to highlight that this property is only used in addressing the second-order term in optimization.
>
> First, the convergence rate is sensitive to the noise from every aspect of the optimization procedure. Reducing the communication cost unavoidably introduces **larger noise**, the norm of which is in-expectation **linear** to $d$. Our analysis manages to address the possible conflicts in the existing analysis of adaptive distributed deep learning by demonstrating that SADL can achieve both sufficient descent (Lemma 2.5) and that the aggregate sketching noise remains bounded (Lemma 2.6). Both terms arise from the first-order component in optimization and thus does not depend on the intrinsic dimensionality.
>
> Second, the current analysis is restricted to un-biased gradient compressors. A vast majority of the existing works (CDAdam, FetchSGD, CAMS etc.) rely on **biased gradient compressors** and error feedback mechanism. The dependence on $d$ in their convergence bound originates from the distortion term (Section C.1 in [1]), where the intrinsic dimensionality is not directly relevant.
>
> - **Q4: Discussion of baseline algorithms.**
>
> We thank the reviewer for the advice. We have included Table 7 in Appendix that summarizes the communication cost and convergence bounds of each baseline algorithm. We have also discussed and categorized the baselines in Appendix A. We will add a more detailed discussion in the final version of this paper.
>
> - **Q5: The third source of noise.**
>
> The third source of noise is essentially a temporal aggregation error which exists in practice. Accumulating stochastic noise naturally gives you an $O(\sqrt{T})$ error by Azuma's inequality.
>
> - **Q6: Typos**
>
> We appreciate the reviewer’s attention in pointing out the typos. We will carefully review the paper and make multiple passes to correct all identified and remaining typos in the final version.
>
> We thank you again for providing the constructive reviews to our work. We hope our responses have provided clarity and addressed your concerns. We look forward to engaging with you in the discussion period.
>
> [1] Richtárik, Peter, Igor Sokolov, and Ilyas Fatkhullin. "EF21: A new, simpler, theoretically better, and practically faster error feedback." Advances in Neural Information Processing Systems 34 (2021): 4384-4396.

---

> > ### Author Response · Authors · 2025-08-07
> >
> > Dear Reviewer UgYH,
> >
> > We thank you again for your valuable feedback and your endorsement of our work. Since the author-reviewer discussion window is coming to a close, we would like to check whether our responses have addressed your questions. Please feel free to let us know if you have any further questions—we would be happy to clarify them.
> >
> > Submission 23560 Authors

---

### Official Review · Reviewer_iDpm · 2025-06-29

**Clarity:** 3
**Significance:** 4
**Originality:** 4
**Rating:** 4
**Confidence:** 3

**Summary:**

In this paper, authors propose the Sketched Adaptive Distributed Learning (SADL) algorithm to combine gradient sketching with adaptive optimizers for reduced communication overhead and faster convergence. The main contributions lie in that authors provide rigorous theoretical proof on the convergence guarantee of the proposed method. Such proof does not rely on previously commonly used assumption of bounded gradient norm; instead, they prove that gradients are implicitly bounded along the optimization trajectory. In addition, they explicitly consider the impact of three types of noises, namely local mini-batch training, sketch-related compression error, and aggregated noise over training horizon. Finally, authors demonstrate the effectiveness of the proposed SADL algorithm for optimizing ResNet, ViT, and BERT models of 40-100 million parameters.

**Questions:**

1. Section 4, line 303. It remains unclear which gradient sketching operator (i.i.d. isotrpoic Gaussian, subsampled randomized Hadamard transform, and count-sketch) and adaptive optimization algorithm is used by SADL in these experiments.
2. Section 4. The ablation study on different combinations of gradient sketching operators and adaptive optimization algorithms is missing. What is the recommended combinations between these two componenets?
3. Is it possible to further extend the proposed method into asynchronous distributed learning setting? This will be significant different from the current setup, and is well out-of-the-scope of this paper, but some brief discussion on this direction might be valuable in the rebuttal.

**Ethical Concerns:**

["NO or VERY MINOR ethics concerns only"]

**Final Justification:**

The proposed SADL algorithm combines gradient sketching with adaptive optimization algorithm. This is more closely related to real-world distributed learning scenarios, where adaptive optimizers rather than vanilla SGD are adopted. This also represents a huge improvement compared to other gradient compression methods that build their convergence analyses based on vanilla SGD and/or in-expectation analysis. The proposed method works well even with extremenly low communication budget, and there is no convergence issue under any sketch size as reported in the experiments. Authors have also addressed my previous concerns in the rebuttal.

**Limitations:**

The empirical results section may be further improved to include detailed ablation study on different combinations of gradient sketching operators and adaptive optimization algorithms.

**Paper Formatting Concerns:**

None.

**Quality:**

3

**Strengths And Weaknesses:**

Strengths:
* The proposed SADL algorithm combines gradient sketching with adaptive optimization algorithm. This is more closely related to real-world distributed learning scenarios, where adaptive optimizers rather than vanilla SGD are adopted. This also represents a huge improvement compared to other gradient compression methods that build their convergence analyses based on vanilla SGD.
* Instead of providing in-expectation analysis of different sources of noise (stochastic gradient, compression error, and sequential dependencies), authors provide high probability convergence bounds as a more convincing guarantee of stable training in practice.
* The convergence rate of the proposed SADL algorithm depends over logarithmically on the ambient dimension $d$ (number of model parameters), which improves over previous results of linear dependency on the ambient dimension. This is established upon the intrinsic dimension of the loss Hessian, which is significantlly smaller than the ambient dimension as proved in several related researches.
* The proposed method works well even with extremenly low communication budget, and there is no convergence issue under any sketch size as reported in the experiments.

Weaknesses:
* Limited experimental results for larger models. Since the bottleneck effect of communcation overhead in distributed learning is more severe for models with larger number of parameters, e.g., billion-scale, it would be nice to see whether the proposed algorithm also works well for such models. However, considering the limited time and possibly insufficient computation resources, it is totally fine to not include such results in the rebuttal.

Notes: I did not fully go through all the detailed theoretical proofs in the appendix, and may not be qualified to justify the correctness of these proofs.

---

> ### Author Rebuttal · Authors · 2025-07-31
>
> We would like to extend our sincere appreciation to the reviewer for all the constructive and positive feedback on our contribution. We are especially grateful that the reviewer acknowledges technical strength of our work in both theory and experiments. We would like to provide detailed responses regarding the weakness part and questions.
>
> - **Q1: Choice of sketching algorithms and optimizers.**
>
> In all experiments, we choose the combination of Adam as the server optimizer and SRHT as the sketching algorithm. Isotropic Gaussian requires creating a dense matrix of size $b \times d$, where $b$ is the sketch size and $d$ is the ambient dimension, which is not memory-efficient, so we do not use that.
>
> We extend our experimental results under the combination of [Adam, AMSGrad] $\times$ [SRHT, CountSketch] and we present the ablation studies for the ResNet experiment in the following tables. The table presents the validation error during the last communication round.
>
> When the communcation cost per-round is $1.6 \times 10^5$ Bytes/epoch,
> |          | SRHT | CountSketch |
> |----------|----------|----------|
> | Adam     | $17.0$%   | $23.8$% |
> | AMSGrad   | $18.1$%   | $25.1$% |
>
> When the communcation cost per-round is $1.6 \times 10^6$ Bytes/epoch,
> |          |  SRHT | CountSketch  |
> |----------|----------|----------|
> | Adam     | $16.7$%   | $23.6$% |
> | AMSGrad   |  $17.0$% | $24.7$% |
>
> Regarding the choice of server optimizer, Adam achieves slightly better performance than AMSGrad. Regarding the choice of sketching methods, SRHT consistently outperforms CountSketch, while the latter is still competitive with other baseline methods in Fig. 1. This observation aligns with the error bound of CountSketch (Lemma B.3), which differs by a $1/\sqrt{b}$ factor with SRHT (Lemma B.1). In summary, Adam + SRHT will be an ideal combination in practice.
>
>
> - **Q2: Extension to larger models and asynchronous training paradigms.**
>
> We appreciate the thoughtful suggestion on the possible extension of our work. Deploying SADL in the wild is definitely an important long-term goal for us.
>
> First, we would like to extend our experimental settings to larger models, as suggested by the reviewer. However, several baseline methods that involve error feedbacks face OOM (out of memory) issues. This is because error feedback requires storing error terms which are of the same size as the full model. We finetune a Deepseek-R1-coder model (1.3B parameters) on the Spider dataset (a complex text-to-SQL generation task) and report the validation accuracy in the following table. We select $b=10^7$, which amounts to 1% compression rate. The only two algorithms that survive the experimental settings are SADL and MARINA. We can observe that SADL outperforms and shows high potentials in the compatibility with large models.
>
> |          |  Accuracy |
> |----------|----------|
> | SADL     |  54.7% |
> | MARINA   |  48.4% |
>
>
> Second, mechanism-wise, SADL naturally extends to asynchronous settings. Our SADL framework can be combined with the generic asynchronous training algorithms such as the one described in Algorithm 1 of [1]. Specifically, lines 3 and 6 in the algorithm block can be modified to incorporate our sketching and desketching operators respectively so that the stochastic gradient is compressed before the transmission. Developing a theoretical analysis on such algorithm will be an interesting extension of our current work.
>
> We thank you again for providing the constructive reviews to our work. We hope our responses have provided clarity and addressed your concerns. We look forward to engaging with you in the discussion period.
>
> [1] Stich, Sebastian, Amirkeivan Mohtashami, and Martin Jaggi. "Critical parameters for scalable distributed learning with large batches and asynchronous updates." International Conference on Artificial Intelligence and Statistics. PMLR, 2021.

---

> ### Author Response · Authors · 2025-08-07
>
> Dear Reviewer iDpm,
>
> We thank you again for your valuable feedback and your endorsement of our work. We hope our response has addressed your questions and further strengthened your confidence in the contributions of our work. Please feel free to let us know if you have any further questions—we would be happy to clarify them.
>
> Submission 23560 Authors

---

> > ### Comment · Reviewer_iDpm · 2025-08-08
> >
> > Dear authors,
> >
> > Thanks for additional experimental results presented in the rebuttal. Results on different combinations of sketching algorithms and optimizers further verified the flexibility of the proposed method. All of my previous concerns have been addressed and I do not have any futher questions.

---

### Official Review · Reviewer_FZNX · 2025-06-29

**Clarity:** 3
**Significance:** 2
**Originality:** 3
**Rating:** 4
**Confidence:** 3

**Summary:**

The authors introduce Sketched Adaptive Distributed Learning (SADL) algorithms. SADL algorithms operate in a distributed learning setting with C clients, where AMSGrad is the outer optimizer, SGD is the inner optimizer, and a sketching operator is used to reduce the size of tensors communicated. The authors provide a new sharp analysis of convergence for SADL with high probability convergence guarantees. The authors find that a sketching dimension logarithmic in the number of model parameters yields a sub-linear convergence rate that only depends on the loss Hessian’s intrinsic dimension. The authors also show that their analysis holds without assuming that gradient norms are bounded, unlike previous work. Finally, the authors provide empirical experiments showing the performance of SADL when compared to other communication-efficient optimizers.

**Questions:**

- Since SOTA models are typically trained with gradient clipping, assuming that gradients are bounded seems realistic. Could you comment on this and explain why the results in section 3 are still important?
- How many Hyperparameters were swept for each method presented in figures 1,2, and 3?
- Are all methods communication-bandwidth matched in figures 1, 2, and 3? For example, sketching is ring all-reduce compatible, while top-k sparsification requires an all-gather.

**Ethical Concerns:**

["NO or VERY MINOR ethics concerns only"]

**Final Justification:**

I have decided to raise by score as the authors have addressed many of my concerns during the rebuttal phase. I recommend accepting the paper, with the caveat that my understanding of the author's theoretical contributions within the context of the existing literature is limited (e.g., the AC should rely on other reviewers for that).

*Issues resolved*
- reporting the hyperparameters that were swept and ensuring a fair HP budget across methods
- the addition of error bars
- results with AMSGrad
- Clarification of AMSGrad or Adam used in the emprirical study
- Clarification about gradient clipping

**Limitations:**

Yes, for the theoretical portion of the paper, but there are some limitations that I highlight in the weaknesses that are not currently addressed in the paper.

**Paper Formatting Concerns:**

**Typos found**:
- Line 32 Folklores → Folklore
- Line 108 dimensioanlity → dimensionality
- Line 128 has a white proof square

**Quality:**

3

**Strengths And Weaknesses:**

**Strengths**:
- The paper does a good job introducing the different theoretical results, placing them within the context of the existing literature, and explaining what is new about the theory shown.
- The theoretical analysis is novel.
- To the extend of my limited understanding of convergence analysis proofs (I am not intimately familiar with the proofs from existing work, but I did read through the math in the main text), the author’s theorems seem sound and to the best of my knowledge, they support what is mentioned in the implications sections/contributions.
- The empirical evaluation covers both image and language tasks with transformers, a key setting to evaluate Adaptive optimizers.

**Weaknesses**:
- **Hyperparameters** While I was able to find the different hyperparameter values used in the appendix, I could not find the number of hyperparameter configurations swept for each optimizer. Given the stochasticity of deep learning, it is crucial to report the hyperparameters swept for each method to demonstrate the performance of different methods under comparable HP budgets.

- **Handling randomness** Following from my first concern above, I find that none of the loss/error curves report error bars. In question 7 of the checklist, the authors mention that “All random seeds are fixed and shared by all experiments”. Simply matching random seeds is not sufficient to control for error in empirical experiments. For instance, a particular method could perform outstandingly better under a particular random initialization and data ordering. I believe that averaging across multiple random seeds is necessary to accurately report the performance of the method.

- **Missing baseline** While AMSGrad was introduced by [1] to ensure the convergence of Adam, the algorithm is rarely used in practice since Adam has less memory overhead and converges when appropriately tuned. I believe this might also be the case for SADL, so including a baseline with Adam as the outer optimizer in the empirical experiments (e.g. a FedAdam [2] baseline) with sketching makes sense and would be beneficial for practitioners. This would also help address my final concern, especially if it can be shown to perform similarly to SADL.

- **misleading writing** The authors claim to introduce “a family of Sketched Adaptive Distributed Learning (SADL) algorithms which can use suitable unbiased gradient sketching for compression with suitable adaptive optimization algorithms” (9-11). This sentence seems misleading as it makes it seem like SADL is broader than a single outer optimizer. However, my understanding is that results have been shown only in the case of an AMSGrad outer optimizer.

---
**References**

[1] [ON THE CONVERGENCE OF ADAM AND BEYOND; ICLR 2018]

[2] [Adaptive Federated Optimization; NeurIPS 2021]

---

> ### Author Rebuttal · Authors · 2025-07-31
>
> We sincerely thank the reviewer for taking the time to provide a detailed and thoughtful review of our paper. We appreciate all the positive comments regarding our contributions, and are especially grateful that the reviewer acknowledges the novelty and soundness of our thoery works. We would like to provide detailed responses regarding the weakness part and questions.
>
> - **Q1: Hyperparameters.**
>
> We explore a wide range of hyperparameters in our experiments. Specifically, we sweep the server learning rate over the set $[10^{-4}, 5\times 10^{-4}, 10^{-3}, 5 \times 10^{-3}, 10^{-2}, 5 \times 10^{-2}, 0.1, 0.5, 1.0, 2.0]$. The algorithms share all the other hyper-parameters within each set of experiments. The only exception is CocktailSGD since it's unclear  how to incorporate server-side momentum in their framework. To ensure a fair comparison, we tune both server learning rate and the client side momentum for CocktailSGD.
>
> - **Q2: Handling randomness.**
>
> We observe that virtually all algorithms, including SADL and the baselines, **exhibit stable behavior** during training. We report the statistical significance level of the ResNet experiment in the following table. Each experiment is repeated three times under different random seeds, with sources of randomness including client data partitioning, data shuffling, and sketching. The table presents the validation error during the last communication round, with the values following $\pm$ indicating $3\times$standard deviation across the runs. 1bit-Adam and PAQ has a fixed compression rate, which amounts to $1.6 \times 10^6$ Bytes/epoch.
>
> | Bytes / epoch         | $1.6 \times 10^5$ | $1.6 \times 10^6$ |
> |----------|----------|----------|
> | **SADL**       | $( 17.8 \pm 2.0)$%   | $(16.9\pm 0.8)$%   |
> | CDAdam   | $(27.7 \pm 3.1)$%   | $(25.4\pm 1.9)$% |
> | CAMS   | $(41.2 \pm 4.9)$%   | $(26.8\pm 1.0)$%   |
> | CocktailSGD   | $(47.4\pm 2.3)$%   | $(31.1\pm 1.7)$%   |
> | MARINA   | $(24.4\pm 2.8)$%   | $(24.8\pm 4.1)$%  |
> | FetchSGD   | $(36.2\pm 2.4)$%   | $(28.2\pm 3.1)$%  |
> | 1bit-Adam   | N/A   |  $(32.3\pm 16.8)$%  |
> | PAQ   | N/A  |$( 24.6\pm 1.7)$%  |
>
>
> - **Q3: "misleading writing" and AMSGrad vs Adam.**
>
> We agree with the reviewer that Adam has been more popular than AMSGrad in practice, and that actually motivates us to choose **Adam as the server optimizer** in all experiments. We will highlight this aspect in the main paper.
>
> For completeness, we provide the experimental results of running AMSGrad on all three experimental settings in the paper. The following table presents the validation error during the last communication round, with the values following $\pm$ indicating $3\times$standard deviation across three independent runs. We can observe that Adam and AMSGrad has comparable performance on all experimental settings, indicating that SADL is insensitive to the choice of optimizers.
>
> |          | ResNet ($1.6 \times 10^5$) | ResNet ($1.6 \times 10^6$) |
> |----------|----------|----------|
> | Adam     | $(17.8 \pm 2.0)$%   | $(16.9 \pm 0.8)$%  |
> | AMSGrad   | $(18.6 \pm 1.5)$%   | $(17.3 \pm 2.1)$%  |
>
> |          | ViT ($3 \times 10^5$) | ViT ($3 \times 10^6$) |
> |----------|----------|----------|
> | Adam     |  $(2.1 \pm 0.1)$%   | $(1.6 \pm 0.1)$%  |
> | AMSGrad   | $(1.8 \pm 0.3)$%   | $(1.6\pm 0.3)$%  |
>
>
> |          | SST ($8 \times 10^5$) | SST ($8 \times 10^6$) |
> |----------|----------|----------|
> | Adam     | $(8.7 \pm 0.2)$%   | $(8.1\pm 0.3)$%  |
> | AMSGrad   | $(8.8 \pm 0.2)$%   |$(8.7\pm 0.3)$%  |
>
> In addition, since the mainstream adaptive optimizers share similar designs in utilizing momentum and preconditioners, the theory we developed can also be easily extended to Adam and other adaptive optimizers.
>
> - **Q4: Avoiding bounded gradient assumption and gradient clipping.**
>
> We agree that the widely used gradient clipping methods guarantee that the gradients are bounded. However, we incorporate the analysis in Section 3 without assuming bounded gradients or using gradient clipping for the following reasons:
> 1. Establishing the convergence result without assuming bounded gradients arguably **broadens the applicability** of SADL, particularly for the cases when the objective function is squared loss, which naturally leads to unbounded gradients.
> 2. We consider the derivation in Section 3 to be a *technically novel* and **potentially valuable contribution**. We believe it may be of independent interest to the broader optimization and machine learning community.
> 3. While gradient clipping effectively addresses the issue of unbounded gradients in practice, it also **introduces bias** to the gradients, which might pose challenges in the theoretical analysis. Understanding and bounding such clipping-induced bias is an important direction of research, especially for the differential privacy community.
>
> - **Q5: Communication bandwidth.**
>
> We thank the reviewer for the insightful comment. In this optimization focused work, we adopt the classical framework where a central parameter server aggregates local updates, as in MARINA[1]. Therefore, our experimental results (Figure 1-3) align the number of parameters (in bytes) that each worker transmits to the server. This alignment protocol follows common practices in prior literature [2, 3]. Furthermore, the adopted framework is general and can be seamlessly applied to federated learning scenarios where direct peer-to-peer communication is often infeasible.
>
> We acknowledge that SADL can benefit from ring all-reduce to mitigate network contentions and improve bandwidth utilization. Practical deployment of SADL will depend on the specific system setting and is an important consideration for industrial applications.
>
> We thank you again for providing the constructive reviews to our work. We hope our responses have provided clarity and addressed your concerns. We look forward to engaging with you in the discussion period.
>
> [1] Gorbunov, Eduard, et al. "MARINA: Faster non-convex distributed learning with compression." International Conference on Machine Learning. PMLR, 2021.
>
> [2] Stich, Sebastian U. "Local SGD converges fast and communicates little." arXiv preprint arXiv:1805.09767 (2018).
>
> [3] Reddi, Sashank, et al. "Adaptive federated optimization." arXiv preprint arXiv:2003.00295 (2020).

---

> > ### Author Response · Authors · 2025-08-03
> >
> > Dear Reviewer FZNX,
> >
> > We have provided our response and hope it addresses all your concerns regarding our work. We sincerely invite you to join the discussion and welcome your valuable feedback. Thank you for your timely thoughts and comments—we truly appreciate them!
> >
> > Submission 23560 Authors

---

> > > ### Comment · Reviewer_FZNX · 2025-08-04
> > >
> > > Thank you for your thorough reply.
> > >
> > > I still have some questions:
> > > - **Tuning inner learning rates** Did you tune the inner optimizer's learning rate for each method? Your reply seems to indicate that you have not when you state "the algorithms share all the other hyper-parameters within each set of experiments." Using the same inner learning rate for all experiments could unnecessarily advantage certain methods over others. How do you think this affects your results?
> > > - **Adam and AMSGrad** Just to clarify, despite algorithm 2 showing the AMSGrad optimizer for ADA_OPT, in all your experiments, you use Adam as the outer optimizer instead?

---

> > > > ### Author Response · Authors · 2025-08-05
> > > >
> > > > We appreciate the reviewer's engagement in the discussion period. We provide the following response to your questions.
> > > >
> > > > - **Q1: Tuning inner learning rates.**
> > > >
> > > > We tune the inner learning rate at the preliminary stage of our work. Specifically, we pick the learning rate within the set $[0.001, 0.01, 0.1, 1.0]$ such that the model can achieve the best validation set performance in a standard centralized training paradigm. Then we adopt the acquired values as the inner learning rates in the following distributed learning frameworks.
> > > >
> > > > This strategy better simulates the practical scenario since the client may be agnostic to the server's hyper-parameters and can only choose the learning rate based on their own training data.
> > > >
> > > > To best accommodate your request on tuning inner learning rates, we have added the following experiments that sweep over both the outer learning rates and inner learning rates. We run under the same setting as the ResNet experiment with communication $1.6 \times 10^5$ Bytes/epoch, and report the best validation error in the following table. The budget of outer learning rates remains the same as in our previous response. The inner learning rates are chosen from the set $[1.0, 0.1, 0.01]$, considering the original inner learning rate is fixed as $0.1$.
> > > >
> > > > | Inner LR       | $1.0$ | $0.1$ | $0.01$ |
> > > > |----------|----------|----------|----------|
> > > > | **SADL**       | $18.9$%   | $17.0$%   | $17.1$% |
> > > > | CDAdam   | $28.0$%   | $27.7$% | $26.9$% |
> > > > | CAMS   | $73.4$%   | $39.3$%   | $42.2$%|
> > > > | CocktailSGD   | N/A  | $47.5$%   | $47.2$% |
> > > > | MARINA   | $30.1$%   | $24.6$%  |  $24.8$%|
> > > > | FetchSGD   | $56.7$%   | $35.4$%  | $44.7$% |
> > > >
> > > > The N/A in CocktailSGD means under the specific inner learning rate, CocktailSGD fails to converge under any choice of the outer learning rate. From these new results, we can observe that for most algorithms, after sweeping the combination over outer (server) learning rates and inner (local) learning rates, yield comparable performance as we originally reported.
> > > >
> > > > We hope the reviewer understands and appreciates the following point: While one can in principle investigate all exponential combinations of hyper-parameters for the experiments, we have done enough to support the proposed SADL, including the new experiments during the past weeks. Also, we would like to highlight the major contribution of the work lies in the theory advances.
> > > >
> > > > - **Q2: Adam and AMSGrad**
> > > >
> > > > You are correct that Adam is used as the outer optimizer in all experiments. This choice also facilitates a fair comparison because all baseline adaptive distributed learning methods adopt Adam as the outer optimizer. We will add the new experimental results of AMSGrad (shown in our previous response) to the paper for completeness, and also highlight the usage of Adam in the experiment section to prevent confusion.
> > > >
> > > > We sincerely thank you for the engagement and providing the valuable feedback in the review process. We hope our response has addressed all your questions.

---

> > > > > ### Comment · Reviewer_FZNX · 2025-08-07
> > > > >
> > > > > Thanks for the response. You have addressed all my concerns.

---

> > > > > > ### Author Response · Authors · 2025-08-07
> > > > > >
> > > > > > Dear Reviewer FZNX,
> > > > > >
> > > > > > We thank you again for your valuable feedback. We are glad to hear that our responses have addressed all your concerns. We hope this has further strengthened your confidence in the contributions of our work, and we would greatly appreciate it if you would consider raising your rating.
> > > > > >
> > > > > > Submission 23560 Authors

---

### Official Review · Reviewer_Dwby · 2025-07-07

**Clarity:** 3
**Significance:** 2
**Originality:** 2
**Rating:** 3
**Confidence:** 3

**Summary:**

This paper studies the problem of distributed optimization with adaptive optimizers applied over compressed local updates from local workers. There are multiple sources of errors that the authors take into account: stochasticity of the problem, extra noise from sketched updates, and the local nature of training. With all of them considered, the authors derive convergence guarantees in the nonconvex setting and study the obtained method empirically on small vision and language tasks.

**Questions:**

"Therefore, SADL can achieve an $O(1/T)$ convergence for small $T$, which yields faster convergence rate than non-adaptive methods."
I don't understand this claim. The right-hand side includes terms that scale as $\sqrt{T}$ and $T$, how exactly would this give $O(1/T)$ convergence rate. How do the authors obtain $O(1/T)$ result?

**Ethical Concerns:**

["NO or VERY MINOR ethics concerns only"]

**Final Justification:**

I read the authors' feedback and the other review. As a primarily theoretical work, I think it suffers from several issues, especially in terms of complexity, which is why I remain on the reject side.

**Limitations:**

yes

**Quality:**

3

**Strengths And Weaknesses:**

The paper's main strength is the technical depth of the study. The authors took a hard problem, derived convergence guarantees, and performed numerical experiments to show how the method works in practice.
It is also worth mentioning that the authors tried their best to provide the most rigorous theoretical results and explained well how one can avoid making any assumptions on boundedness of the gradients or iterates.

The main weakness of the work is, in my opinion, limited motivation for all the complexity. The adaptive methods are not well understood (e.g. no clean theory exists to explain what exactly makes rmsprop/adam so good) and their theory isn't particularly attractive for extending to more complicated settings. Most other components such as local updates, compression, and momentum are well understood, but it still remains unclear how much insight we get from combining them all together.
Overall, this paper seems to suggest that the combination makes sense to try to use and see if it works well, but this is often the case already with different optimizers: we need to try them in practice to know if they work even if there is theory supporting them. That could be studied by the authors, but their numerical evaluations are rather limited. A good example of how a study like this could be done is the recent work of Douillard et al. "Streaming DiLoCo with overlapping communication: Towards a Distributed Free Lunch", which the authors seem to now know, but it provides state of the art in distributed optimization.

Minor things to fix:
Typo in line 78: "in both the the first moments"
Line 206: "left hand side" -> "left-hand side"

---

> ### Author Rebuttal · Authors · 2025-07-31
>
> We sincerely thank the reviewer for taking the time to provide a detailed and thoughtful review of our paper. We appreciate all the positive comments regarding our contributions, and are especially grateful that the reviewer acknowledges the technical depth of our work. We would like to provide detailed responses regarding the weakness part and questions.
>
> - **Q1: "limited motivation for all the complexity"**
>
> While the SADL algorithm, as described in Alg. 1, is simple and concrete, the analysis is indeed involved since introducing sketching into adaptive distributed optimization might cause conflicts in various components of optimization, such as the preconditioning and the additional source of sketching noise. The **key insight** of our work is that both the convergence rate and the communication cost of SADL depends only **logarithmically** on the ambient dimension $d$, which has an exponential improvement over the existing bound which is **linear** in $d$. The result benefits from our novel analysis on the terms that could be directly affected by the use of sketching.
>
> 1. The convergence rate is sensitive to the noise from every aspects of the optimization procedure. Reducing the communication cost unavoidably introduces **larger noise**, the norm of which is in-expectation **linear** to $d$. Our analysis manages to address the possible conflicts in the existing analysis of adaptive distributed deep learning by demonstrating that SADL can achieve both sufficient descent (Lemma 2.3 and 2.5) and that the aggregate sketching noise remains bounded (Lemma 2.6).
> 2. Second, the use of adaptive optimizers poses unique challenges in leveraging the **anisotropic** structure (Definition 2.1) of deep learning optimization in the analysis. Adaptive optimizers naturally re-scales the update step to be **isotropic** over the ambient dimension via preconditioning. Our sharp analysis shows that the adaptive steps preserve the structure of the sketching noise so that the variance term can be bounded (Lemma 2.4).
>
> In summary, our work provides the _first theoretical analysis_ that **sketching is compatible with all components of off-the-shelf adaptive distributed deep learning** methods. Each term in the convergence bound depends at most logarithmically on $d$ and we are the first to show this key insight.
>
> - **Q2: "numerical evaluations are rather limited"**
>
> We would like to emphasize that our work primarily focuses on _distributed deep learning_ **theory**. The numerical experiments are included to illustrate and support our novel theoretical findings.
>
> We appreciate the reviewer's pointer to [1]. Our main focus is on reducing the communication cost while preserving convergence guarantees. In comparison, [1] considers other aspects of distributed learning, such as synchronization mechanisms, which fall outside the scope of our paper. The experiment in [1] features simulations of compute utilization metric with varying bandwidths. However, such experimental setting is not suitable to demonstrate our theoretical findings. Since our algorithm and all baseline methods share the same synchronization mechanism, the effect of network bandwidth impacts all methods equally when the per-round communication cost is the same.
>
> Papers like [1] are highly sophisticated for real-world deployment and involve various building blocks, while we are more focused on the core distributed learning algorithm with reduced communication. We believe that our proposed **SADL can serve as a foundational component** for future work, including works such as [1].
>
> - **Q3: SADL's $O(1/T)$ convergence for small $T$.**
>
> The statement is made from Corollary 2 for _small $T$_, i.e., $T \le O(1/\epsilon^2)$, and the right-hand-side of the bound only involves $O(1/T)$ terms.
>
> We thank you again for providing the constructive reviews to our work. We hope our responses have provided clarity and addressed your concerns. We look forward to engaging with you in the discussion period.
>
> [1] Douillard, Arthur, et al. "Streaming diloco with overlapping communication: Towards a distributed free lunch." arXiv preprint arXiv:2501.18512 (2025).

---

> > ### Author Response · Authors · 2025-08-03
> >
> > Dear Reviewer Dwby,
> >
> > We have provided our response and hope it addresses all your concerns regarding our work. We sincerely invite you to join the discussion and welcome your valuable feedback. Thank you for your timely thoughts and comments—we truly appreciate them!
> >
> > Submission 23560 Authors

---

> > > ### Comment · Reviewer_Dwby · 2025-08-04
> > >
> > > Dear authors,
> > >
> > > Thanks for providing a response to my comments.
> > >
> > > > the analysis is indeed involved since introducing sketching into adaptive distributed optimization might cause conflicts in various components of optimization
> > >
> > > I understand, but I don't think that having a complicated analysis is a strength, as the guarantees become less accessible to the general reader and harder to extend.
> > >
> > > > The key insight of our work is that both the convergence rate and the communication cost of SADL depends only logarithmically on the ambient dimension $d$
> > >
> > > Unless we know that it's optimal, this is still a disappointing result as most bounds in optimizations don't depend on the dimension at all (e.g., the convergence rate of SGD).
> > >
> > > After reading the authors' comments, I decided to keep the initial score I assigned to the paper.

---

> > > > ### Author Response · Authors · 2025-08-05
> > > >
> > > > We really appreciate the reviewer engaging in the discussion period.
> > > >
> > > > > this is still a disappointing result as most bounds in optimizations don't depend on the dimension at all (e.g., the convergence rate of SGD).
> > > >
> > > > You are right that many old optimization results had dimensional dependence (Bubeck's monograph Ch. 2 covers this old material), which modern methods like gradient descent has gotten rid off, a long time back.
> > > >
> > > > It is **critical to understand** that the source of dimensional dependence in modern *communication-efficient* distributed deep learning stems from the need for ***gradient compression***----we have covered some of this literature in Appendix A.
> > > > * The existing communication-efficient distributed deep learning methods pick up a dimensional dependence due to gradient compression, and the dependence is $O(d)$.
> > > > * Our work is the first to get a $O(\log d)$ dependence, which is an exponential improvement----from $O(d)$ to $O(\log d)$----over the existing literature.
> > > > * Our strong empirical performance demonstrates that this is not just some math analysis thing, our SADL is indeed persistently better than these baselines.
> > > >
> > > > The above summarizes our main technical point (which we really hope the reviewer will appreciate). We provide some additional technical remarks, in comparison with SGD:
> > > >
> > > > Technically, the difference in the noise level is the key reason that makes centralized SGD have a different convergence rate -- and that's why we must bring up the  so-called "complexities" in the analysis.
> > > > * Centralized SGD has a dimension-independent noise term (for example, sub-Gaussian noise[1]), while communication-efficient distributed learning generally has a noise *linear in the dimension* because of the compression[2].
> > > > * This difference in the noise level leads to unique challenges in the theoretical analysis of communication-efficient distributed learning.
> > > > *  Our work shows that SADL matches the **convergence rate** of centralized SGD up to a logarithmic factor on $d$ with **communication cost** independent of $d$. For comparison, if we transmit the full parameter in each communication round, although the convergence rate may still be independent of $d$, the total communication will be *linear to $d$*.
> > > >
> > > > Therefore, we feel our result is actually counter-intuitive and pleasantly surprising.
> > > >
> > > > > Unless we know that it's optimal,
> > > >
> > > > While we do not claim the result to be "optimal" (no matching lower bound), our work has accomplished an **exponential improvement** over the existing bounds on the convergence rate, from $O(d)$ to $O(\log d)$, which is a substantial improvement.
> > > >
> > > > > I don't think that having a complicated analysis is a strength, as the guarantees become less accessible to the general reader and harder to extend.
> > > >
> > > > We understand your concern--please let us make two points.
> > > >
> > > > First, we have provided a **high probability** (not in-expectation) bound while handling three different sources of noise, leading to an exponential improvement in dimensional dependence----considering the complexities of the analysis was ***necessary*** to derive our novel bound. Further, the algorithm works very well in practice----so we have a theoretically sound algorithm which works well in practice.
> > > >
> > > > Second, we are trying to **cater to two audiences** in the paper:
> > > > * The primarily empirical audience, who need to clearly understand the SADL algorithm (which we have clearly described) and study the empirical results (which we have clearly presented). We have also provided code along with the submission, so one can verify and build on SADL.
> > > > * The primarily optimization/ML theory audience, who need to understand the gory technical details of the analysis. For this audience, we have provided proof sketches in the main paper, shared various remarks to illustrate the insights from the theorems, and provided detailed proofs in the appendix.
> > > >
> > > > In summary, we have done our best to make the paper accessible to the broad audience, while still maintaining technical depth.
> > > >
> > > > We sincerely thank you for engaging and sharing your thoughts. We sincerely hope that you will read our response and re-assess the contribution of this work.
> > > >
> > > > [1] Harvey, N.J., Liaw, C., Plan, Y. and Randhawa, S., 2019, June. Tight analyses for non-smooth stochastic gradient descent. In Conference on Learning Theory (pp. 1579-1613). PMLR.
> > > >
> > > > [2] Beznosikov, A., Horváth, S., Richtárik, P. and Safaryan, M., 2023. On biased compression for distributed learning. Journal of Machine Learning Research, 24(276), pp.1-50.

---

> > > > ### Author Response · Authors · 2025-08-07
> > > >
> > > > Dear Reviewer Dwby,
> > > >
> > > > We thank you again for engaging in the discussion and providing valuable feedback. Since the author-reviewer discussion window is coming to a close, we would like to check whether our responses have addressed your concerns. Please feel free to let us know if you have any further questions—we would be happy to clarify them.

---

### Comment · Area_Chair_8sM8 · 2025-08-02

Dear Reviewers

The authors have responded to your reviews. In the next few days, please read their responses and engage in a productive discussion that will be critical to the review process.

I truly appreciate your timely thoughts and comments!

AC

---

### Note · Authors · 2025-08-14

Dear AC and Reviewers,

We thank you again for your efforts throughout the review process. We are pleased that Reviewers FZNX and iDpm acknowledged all their concerns being addressed. We hope you will consider the improvements made during the rebuttal period in your final evaluation.

As the final remark, we present a summary on how each concern has been resolved.

On theory aspects,
- **Theoretical Motivation (Dwby Q1, UgYH Q2)**

Our result presents an **exponential improvement** (from $O(d)$ to $O(\log d)$) in the convergence rate over existing communication-efficient distributed learning algorithms. The analysis on adaptive optimization and the three sources of noises are **essential** to derive this counter-intuitive result and ensure the theory's **practical applicability**.
- **Dependence on Assumptions (FZNX Q4, UgYH Q3)**

We explained the connection between our results and the intrinsic dimension, clarified which key steps are independent of intrinsic dimension, and highlighted the technical significance of removing the bounded gradient assumption.
- **Related Works (UgYH Q1, UgYH Q4)**

We discussed the connection and differences with (Song et al. 2023) and provided pointers to our discussion of baseline methods.
- **Clarity in Theoretical Claims (Dwby Q3, UgYH Q5)**

We resolved the misunderstandings of Corollary 2 and clarified the source of the "third noise".

On empirical aspects,
- **Experimental Configuration (Dwby Q2, FZNX Q5, iDpm Q2)**

We justified our experimental setup -- Our setup follows common practices; The experiments are suitable to validate the theoretical claims.
- **Hyper-parameter Tuning (FZNX Q1)**

We provided tuning budgets and tuned client learning rates during rebuttal. **The updated results stay consistent with our main findings** on the efficacy of SADL.
- **Statistical Significance (FZNX Q2)**

We reported statistical significance during rebuttal. SADL exhibits strong stability and consistently better performance.
- **Flexibility of SADL algorithm (FZNX Q3, iDpm Q1)**

We conducted new experiments with different combinations of sketching methods and adaptive optimizers. The results **confirm SADL’s flexibility** and we provide empirical guidance on the combination.

In summary, SADL has a considerably sharper convergence guarantee alongside strong empirical performance. We sincerely hope these contributions will be recognized by the community and serve as a foundation for future research in distributed learning.

---

### Decision · Program_Chairs · 2025-09-17

**Decision:**

Accept (poster)

**Comment:**

This submission combines sketching with adaptive distributed deep learning methods.
The authors improve the convergence bound dependce from d to log d compared with other compression methods. After engaging discussion some reviewers raised their scores. While I agree with reviewer Dwby that the paper could be improved and that the dependen on d is unfortunate, I also think that the improvement from d to log d is sufficiently interesting and recommend acceptance.